# *Drosophila* MIC10b can polymerize into cristae-shaping filaments

Till Stephan[1,2,]*, Stefan Stoldt[1,2,3,]*, Mariam Barbot[1,2,]*, Travis D Carney[4,5], Felix Lange[1,2], Mark Bates[1,6], Peter Bou Dib[1,2], Kaushik Inamdar[1,2], Halyna R Shcherbata[4,5], Michael Meinecke[7], Dietmar Riedel[8], Sven Dennerlein[9], Peter Rehling[3,9,10,11], Stefan Jakobs[1,2,3,10]

Cristae are invaginations of the mitochondrial inner membrane that are crucial for cellular energy metabolism. The formation of cristae requires the presence of a protein complex known as MICOS, which is conserved across eukaryotic species. One of the subunits of this complex, MIC10, is a transmembrane protein that supports cristae formation by oligomerization. In *Drosophila melanogaster*, three MIC10-like proteins with different tissue-specific expression patterns exist. We demonstrate that CG41128/MINOS1b/DmMIC10b is the major MIC10 orthologue in flies. Its loss destabilizes MICOS, disturbs cristae architecture, and reduces the life span and fertility of flies. We show that DmMIC10b has a unique ability to polymerize into bundles of filaments, which can remodel mitochondrial crista membranes. The formation of these filaments relies on conserved glycine and cysteine residues, and can be suppressed by the co-expression of other *Drosophila* MICOS proteins. These findings provide new insights into the regulation of MICOS in flies, and suggest potential mechanisms for the maintenance of mitochondrial ultrastructure.

## Introduction

Mitochondria are organelles that are key for the energy metabolism of all eukaryotic cells. They rely on a double-membrane structure to generate the ubiquitous energy carrier ATP by oxidative phosphorylation. The two mitochondrial membranes exhibit distinct protein and lipid compositions and a strikingly different topology. The smooth outer membrane (OM) surrounds the organelle and mediates the import of metabolites and proteins. The inner membrane (IM) exhibits a much larger surface and is functionally and structurally subdivided into two domains. The inner boundary membrane (IBM) parallels the OM, whereas the crista membranes (CM) form well-organized invaginations, termed cristae, which point toward the interior of the organelle. The cristae are connected to the IBM by small openings, referred to as crista junctions (CJs), which feature a diameter of around 20–30 nm (Mannella et al, 1997). Proper folding of the IM into cristae is closely related to the function of mitochondria, and a disturbed mitochondrial ultrastructure has been associated with human diseases, including neurodegenerative disorders, metabolic diseases, and cardiomyopathies (Chan, 2012; Nunnari & Suomalainen, 2012; Friedman & Nunnari, 2014; Pernas & Scorrano, 2016; Wai & Langer, 2016; Suomalainen & Battersby, 2018).

The mitochondrial contact site and cristae organizing system (MICOS) is a large hetero-oligomeric complex situated in the IM (Harner et al, 2011; Hoppins et al, 2011; von der Malsburg et al, 2011). It is highly conserved across protozoa, fungi, and animals (Wideman & Muñoz-Gómez, 2018) and consists of at least seven different proteins (MIC60, MIC27, MIC26, MIC25, MIC19, MIC13, and MIC10) in humans (van der Laan et al, 2016; Mukherjee et al, 2021). MICOS is a crucial determinant for the biogenesis and maintenance of cristae and CJs. The holo-MICOS complex consists of two subcomplexes, namely, the MIC60 subcomplex and the MIC10 subcomplex. The MIC60 subcomplex, composed of MIC60, MIC25, and MIC19, is critically important for the stability of MICOS and the formation of CJs. Loss of MIC60 causes degradation of all MICOS proteins, disrupts virtually all CJs, and strongly disturbs the cristae organization in yeast and human cells (Harner et al, 2016; Li et al, 2016; Kondadi et al, 2020; Stephan et al, 2020). The MIC10 subcomplex, consisting of MIC10, MIC13, MIC26, and MIC27 in humans, is crucial for a proper

[1]Department of NanoBiophotonics, Max Planck Institute for Multidisciplinary Sciences, Göttingen, Germany   [2]Clinic of Neurology, University Medical Center Göttingen, Göttingen, Germany   [3]Cluster of Excellence "Multiscale Bioimaging: from Molecular Machines to Networks of Excitable Cells" (MBExC), University of Göttingen, Göttingen, Germany   [4]Institute of Cell Biochemistry, Hannover Medical School, Hanover, Germany   [5]Mount Desert Island Biological Laboratory, Bar Harbor, ME, USA   [6]Department of Optical Nanoscopy, Institute for Nanophotonics, Göttingen, Germany   [7]Biochemistry Center (BZH), Heidelberg University, Heidelberg, Germany   [8]Laboratory of Electron Microscopy, Max Planck Institute for Multidisciplinary Science, Göttingen, Germany   [9]Department of Cellular Biochemistry, University Medical Center Göttingen, Göttingen, Germany   [10]Fraunhofer Institute for Translational Medicine and Pharmacology, Translational Neuroinflammation and Automated Microscopy, Göttingen, Germany   [11]Max Planck Institute for Multidisciplinary Science, Göttingen, Germany

Correspondence: sjakobs@gwdg.de
*Till Stephan, Stefan Stoldt, and Mariam Barbot contributed equally to this work

cristae architecture and influences the distribution of CJs. MIC10, a core subunit of this MICOS subcomplex, is a small, highly conserved transmembrane (TM) protein of about 70–100 amino acids, depending on the species. Biochemical studies of the budding yeast ScMIC10 suggest that MIC10 features a hairpin-like structure with its N- and C-termini pointing toward the intermembrane space (IMS). Each of the two TM domains of MIC10 contains a highly conserved glycine-rich motif (Barbot et al, 2015; Bohnert et al, 2015). For ScMIC10, it has been shown that these glycine-rich motifs mediate the formation of stable oligomers that are suggested to bend the mitochondrial IM to support cristae formation (Barbot et al, 2015; Bohnert et al, 2015).

MICOS is best studied in budding yeast and human cells, whereas it is less well characterized in the fly *Drosophila melanogaster*. Depletion of the MIC60 orthologue mitofilin/Dmel_CG6455 results in aberrant cristae morphologies and has been related to impaired synaptic functions, leading to the death of flies in the late pupal stage (Tsai et al, 2017). Likewise, depletion of the MIC13 orthologue QIL1/Dmel_CG760 or of the MIC26-MIC27 orthologue Dmel_CG5903 resulted in altered mitochondrial morphology and aberrant cristae (Guarani et al, 2015; Wang et al, 2020). These morphological changes were associated with reductions in climbing activity, indicating deficits in muscle function (Wang et al, 2020).

Genome sequencing (FlyBase, release FB2022_06) suggested the existence of three MIC10 orthologues in *D. melanogaster* (Dmel_CG12479/MINOS1a, Dmel_CG41128/MINOS1b, and Dmel_CG13564/MINOS1c) (Pfanner et al, 2014; Gramates et al, 2022). Their functional role has so far not been reported in any detail. In this study, we show that the ubiquitously expressed Dmel_CG41128/MINOS1b is the major MIC10 orthologue in flies, which we refer to as DmMIC10b. Loss of DmMIC10b disturbs mitochondrial ultrastructure and reduces the life span of flies. The overexpression of DmMIC10b leads to the formation of long cristae-shaping filaments along the IMS. We demonstrate that this striking behavior of DmMIC10b relies on several conserved amino acid residues and can be efficiently suppressed by the co-expression of DmMIC13 or DmMIC26, but not by their human orthologues. The findings provide new insights into the regulation of MIC10 oligomerization.

## Results

### The mitochondrial protein Dmel_CG41128 is homologous to MIC10 from yeast and humans

The genome of *D. melanogaster* encodes three different proteins with noticeable sequence similarity to the MIC10 proteins from yeast (ScMIC10) and humans (HsMIC10): MINOS1a/Dmel_CG12479; MINOS1b/Dmel_CG41128; and MINOS1c/Dmel_CG13564. Like MIC10 from yeast and humans, these three proteins contain two putative transmembrane domains (TMDs) with conserved glycine-rich motifs (Fig 1A). Each N-terminal TM segment contains a GxxxG motif (Engelman motif), which has been reported to mediate oligomerization of several membrane proteins (Russ & Engelman, 2000).

Each C-terminal TM domain contains a related, highly conserved GxGxGxG motif, which has been shown to be crucial for the oligomerization of ScMIC10 (Barbot et al, 2015; Bohnert et al, 2015).

Of the three orthologues, only Dmel_CG41128 is ubiquitously expressed throughout all analyzed developmental stages, whereas Dmel_CG12479 and Dmel_CG13564 are testes-specific proteins (FlyBase FB2022_06). Therefore, we decided to focus on the investigation of Dmel_CG41128.

The expression of Dmel_CG41128-FLAG in S2 cells and subsequent immunolabeling and fluorescence microscopy demonstrated that Dmel_CG41128 indeed localized to mitochondria (Fig 1B). We next used AlphaFold2 (Jumper et al, 2021) to predict the overall structure of Dmel_CG41128, ScMIC10, and HsMIC10. The AlphaFold2 algorithm predicted a hairpin topology reminiscent of some ER-resident reticulons (Yang & Strittmatter, 2007) for all three proteins, with the conserved GxxxG and GxGxGxG motifs being oriented to each other in a similar way (Fig 1C). The predicted structures are fully in line with previous experimental studies on ScMIC10, which demonstrated that two TM domains of ScMIC10 are linked by a short loop that points toward the matrix, whereas the termini of the protein point toward the IMS (Barbot et al, 2015; Bohnert et al, 2015).

We conclude that Dmel_CG41128 is a mitochondrial protein that shows sequence homology with known MIC10 proteins and presumably features a comparable hairpin-like shape, making it a promising candidate for the MIC10 subunit from *D. melanogaster*.

### Loss of Dmel_CG41128 reduces the life span and the fertility of flies

In humans, the loss of subunits of the MIC10 subcomplex is associated with severe diseases such as mitochondrial encephalopathy, myopathy, or cognitive impairment (Guarani et al, 2016; Zeharia et al, 2016; Benincá et al, 2021). To investigate the influence of Dmel_CG41128 on mitochondria and the life span of flies, we generated flies deficient for Dmel_CG41128 using CRISPR/Cas9 genome editing (Figs 2A and B and S1). The loss of Dmel_CG41128 reduced the average life span of flies by around 40% (Fig S2A). Moreover, the loss of Dmel_CG41128 reduced the fly's overall fertility, indicated by a significantly decreased seminal vesicle area (Fig S2B), which is in line with the finding that Dmel_CG41128 was present in both testes and ovaries from *D. melanogaster* (Fig S3A–H).

### Dmel_CG41128 is the major MIC10 orthologue in *D. melanogaster*

In both humans and yeast, MIC10 controls the stability of the MIC10 subcomplex. Depletion of MIC10 leads to the degradation of MIC13 and strongly disturbs cristae architecture (Harner et al, 2011; Hoppins et al, 2011; von der Malsburg et al, 2011; Alkhaja et al, 2012; Callegari et al, 2019; Kondadi et al, 2020; Stephan et al, 2020). To investigate the consequences of Dmel_CG41128 depletion on mitochondrial architecture in *D. melanogaster*, we isolated mitochondria from WT and Dmel_CG41128-deficient flies and analyzed them by SDS–PAGE and immunoblotting. Similar to the situation in yeast and human cells (von der Malsburg et al, 2011; Guarani et al, 2015; Kondadi et al, 2020), loss of Dmel_CG41128 caused the loss of DmMIC13 when analyzing steady-state levels (Fig 2A). Next, we performed co-immunoprecipitation experiments on the isolated WT mitochondria using Dmel_CG41128 as a

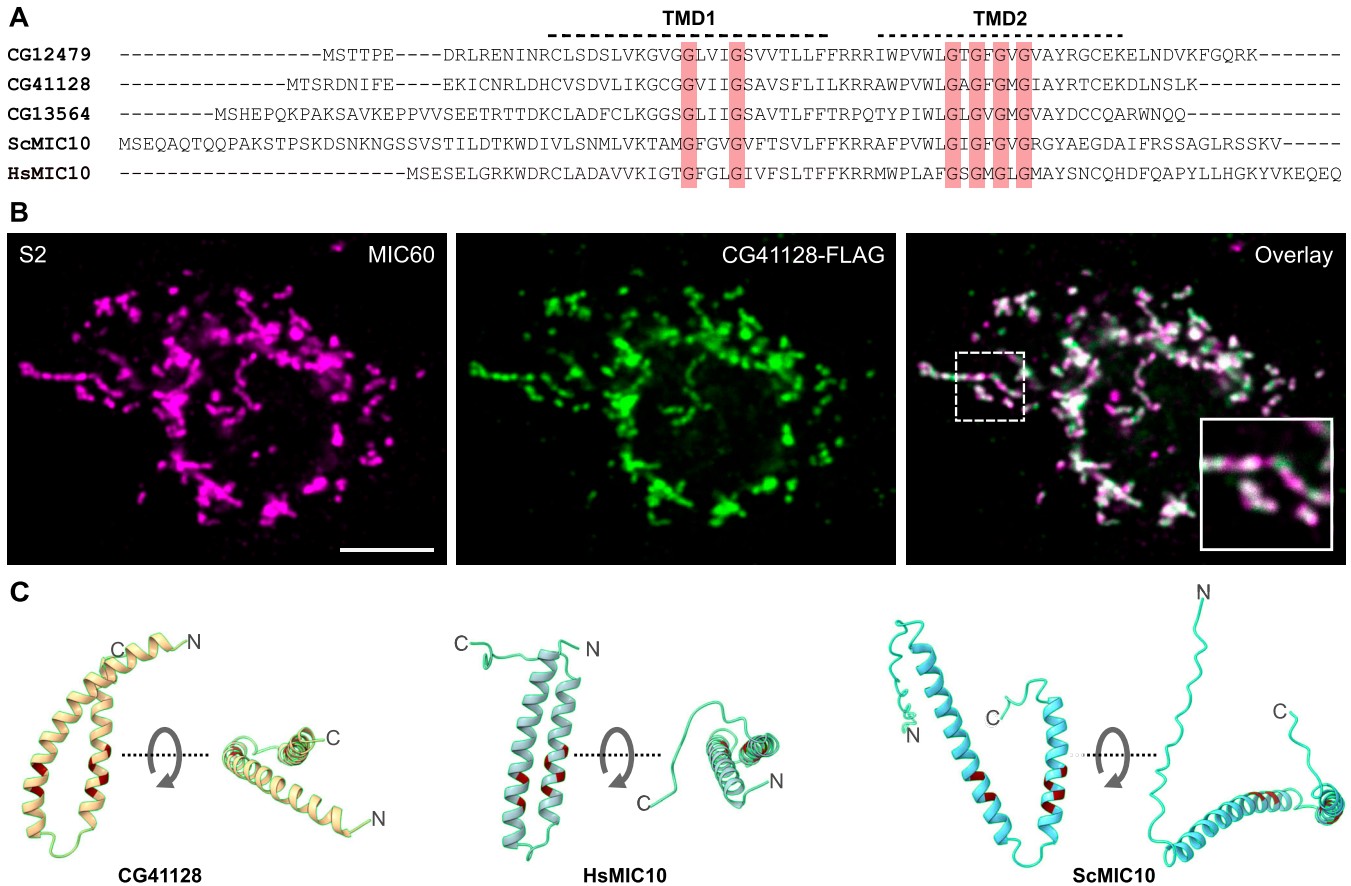

**Figure 1.** ***Drosophila melanogaster* has three MIC10-like proteins.**
**(A)** Sequence alignment of three putative MIC10 proteins from *D. melanogaster*, MIC10 from *S. cerevisiae* (ScMIC10), and MIC10 from *H. sapiens* (HsMIC10). The two putative transmembrane domains (indicated by dashed lines) and the conserved GxxxG and GxGxGxG motifs (red shading) are highlighted. **(B)** Immunofluorescence recording of an S2 cell expressing CG41128-FLAG. **(C)** AlphaFold2 structure predictions of CG41128, HsMIC10, and ScMIC10. Positions of conserved glycine residues are highlighted in red. For each protein, two different views of the protein structure are shown. Scale bar: 5 µm.

bait (Fig 2B). We found that Dmel_CG41128 interacted with both DmMIC13 and DmMIC60, central subunits of both MICOS sub-complexes (Guarani et al, 2015). Likewise, when DmMIC60 was used as a bait, Dmel_CG41128 and DmMIC13 were pulled down (Fig 2B), indicating that Dmel_CG41128 is a part of the *Drosophila* MICOS complex.

To test whether Dmel_CG41128 also interacts with subunits of mammalian MICOS complexes, we expressed Dmel_CG41128-FLAG and FLAG-tagged human MIC10 (HsMIC10-FLAG) in human HeLa cells and COS-7 cells from the green African monkey *Cercopithecus aethiops*. Approximately, the same amounts of MICOS subunits were pulled down with these proteins as baits in COS-7 cells. Also, in HeLa cells Dmel_CG41128-FLAG pulled down subunits of MICOS, although HsMIC10-FLAG was a more efficient bait (Fig 2C). We conclude that Dmel_CG41128-FLAG interacts also with the mammalian MICOS complex.

MIC10-deficient mitochondria from yeast and human cells exhibit highly disturbed mitochondrial ultrastructure (Harner et al, 2011; Hoppins et al, 2011; von der Malsburg et al, 2011; Guarani et al, 2015). Transmission electron microscopy of brain tissue mitochondria of flies deficient for Dmel_CG41128 revealed similar mitochondrial phenotypes with most of the mitochondria exhibiting

aberrant crista morphologies including tube-like and onion-shaped cristae (Fig 2D and E).

Altogether, the ubiquitously expressed Dmel_CG41128 shares sequence homology with other MIC10 proteins, presumably features a hairpin-like structure, and binds to the MICOS complexes of *D. melanogaster*, *H. sapiens*, and *C. aethiops*. Dmel_CG41128 further regulates the levels of DmMIC13 in flies, and its depletion strongly affects the cristae architecture. These findings support the view that Dmel_CG41128 is the ubiquitously expressed MIC10 orthologue in *D. melanogaster*. Hence, in accordance with the uniform nomenclature for MICOS (Pfanner et al, 2014), we will refer to it as DmMIC10b from here on.

## DmMIC10b has a propensity to polymerize into filaments

When analyzing the subcellular localization of overexpressed DmMic10b in S2 fly cells using confocal fluorescence microscopy, it became apparent that at higher expression levels, DmMic10b-FLAG influenced the overall shape of the mitochondrial network (Fig 3A). We next performed super-resolution microscopy to investigate the distribution of DmMIC10b in more detail. At moderate expression

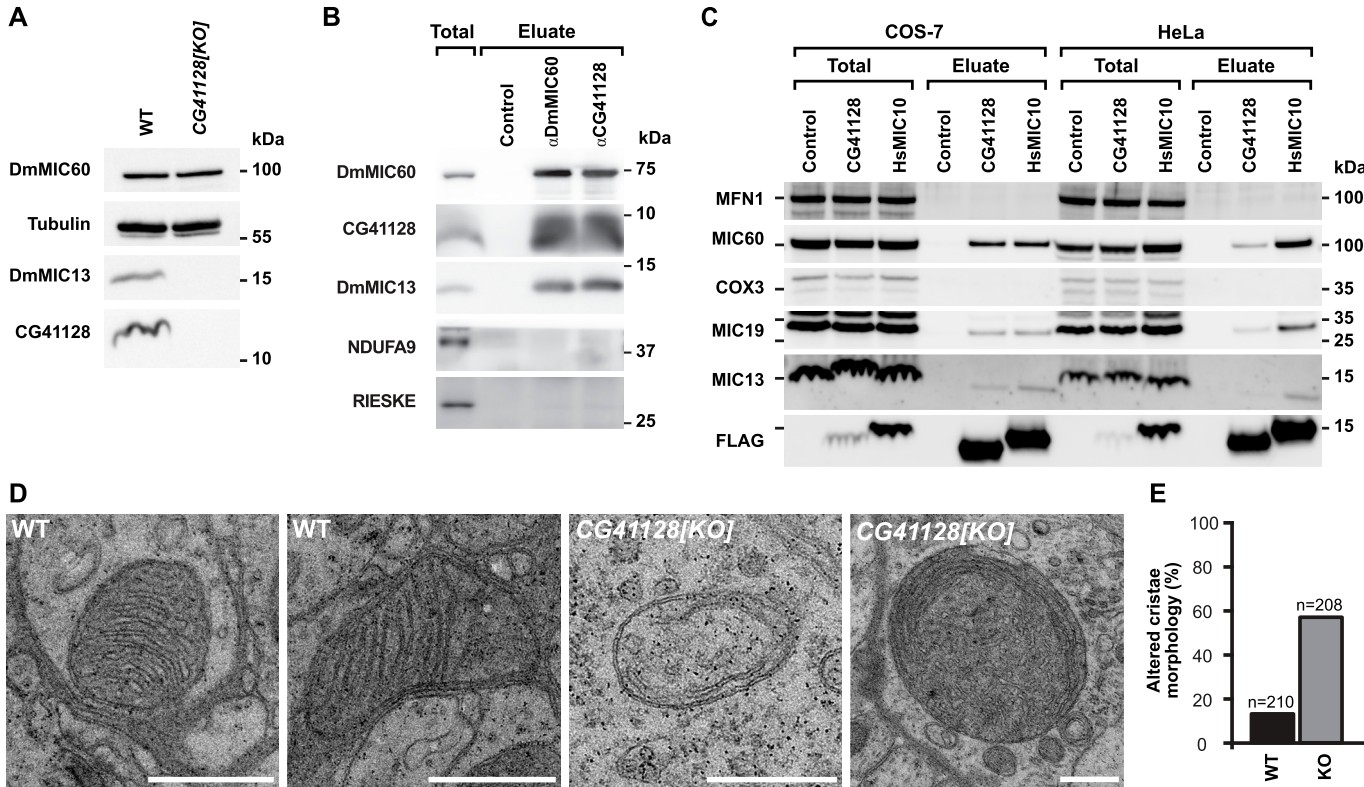

**Figure 2. Dmel_CG41128 is DmMIC10b, a bona fide subunit of MICOS in *D. melanogaster*.**
**(A)** Western blot analysis of steady-state protein levels in solubilized mitochondria from WT and *CG41128[KO]* flies. **(B)** Co-IP from solubilized mitochondria from WT flies. DmMIC60 and CG41128 were used as a bait. **(C)** Co-IP from whole-cell lysates. Human MIC10-FLAG (HsMIC10) and CG41128-FLAG were transiently expressed in HeLa cells or COS-7 cells. FLAG-tagged proteins were used as a bait. **(D, E)** Electron micrographs of mitochondria in brain tissue from WT and *CG41128[KO]* flies. n, number of analyzed mitochondrial cross sections. **(D)** Representative images of mitochondria. **(E)** Quantification of the cristae architecture. Scale bars: 0.5 μm.

levels, STED nanoscopy revealed the formation of distinct DmMIC10b clusters, comparable to the situation in yeast and human cells (Jans et al, 2013). Intriguingly, long DmMIC10b-containing filaments, apparently forming bundles of filaments, were observable at higher expression levels of DmMIC10b-FLAG (Fig 3A and B). These DmMIC10b-containing filaments seemed to be located inside of mitochondria and pervaded the mitochondrial network (Fig 3A–C). Even when scrutinized by STED microscopy, the filaments appeared to be contiguously labeled, suggesting that they might be formed by DmMIC10b only, rather than being a mosaic out of DmMIC10b and other proteins (Fig 3B). To explore this idea, we first expressed DmMIC10b-FLAG in two heterologous cellular systems, namely, HeLa and COS-7 cells. Similar to the overexpression in S2 cells, the expression of DmMIC10b in these cells resulted in the formation of DmMIC10b-containing filaments, which formed exclusively inside mitochondria (Figs 4A and S4A and B).

As DmMIC10b interacted with mammalian MIC60 (Fig 2C), we analyzed the fine distribution of MIC60 in these DmMIC10b-overexpressing cells. Dual-color 2D STED recordings of COS-7 cells indeed suggested occasional spatial connections between the filaments and some MIC60 clusters (Fig 4A). MIC60 seemed to be localized in clusters along the IBM as described before (Harner et al, 2011; Jans et al, 2013; Stoldt et al, 2019; Pape et al, 2020), whereas DmMIC10b-containing filaments often seemed to be situated more toward the center of the mitochondrial tubules (Fig 4A, Inset 1). To

localize DmMIC10b in 3D, we next performed 4Pi-STORM of COS-7 cells expressing DmMIC10b-FLAG (Bates et al, 2022). At low expression levels, DmMIC10b was found in close proximity to MIC60 clusters, suggesting that it was mainly located at CJs or in the IBM (Fig S5A–C). However, the 3D recordings confirmed that at higher expression levels, a large fraction of DmMIC10b was present in filamentous structures (Video 1). Most of these long filaments were located along the center of mitochondrial tubules with no obvious connection to the MIC60 clusters indicating the IBM (Figs 4B and S5A and Video 2).

### Filament formation does not require other MICOS proteins and is independent of the C-terminal tag

The nanoscopy data supported the idea that DmMIC10b-FLAG filaments may form independent from other MICOS proteins. To test this further, we next expressed DmMIC10b-FLAG in human MIC10-KO and MIC60-KO cells, as these cells are devoid of the MIC10 subcomplex and the entire MICOS complex, respectively (Stephan et al, 2020). STED recordings revealed that the formation of DmMIC10b filaments occurred in the absence of the mammalian MIC10 subcomplex or mammalian holo-MICOS complex (Figs 4C and S6A and B). We conclude that additional MICOS subunits are not required for the polymerization of DmMIC10b into filaments.

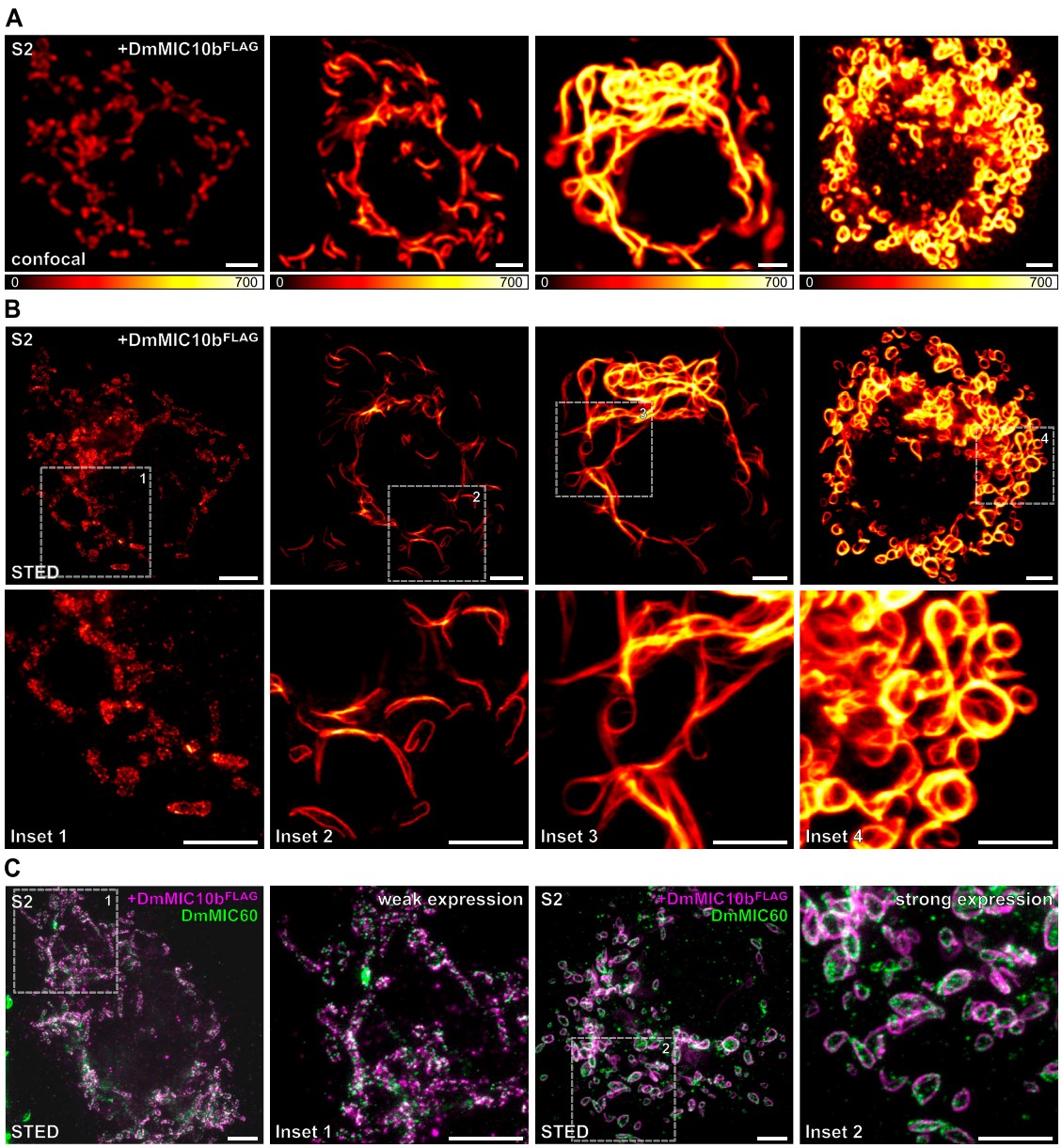

**Figure 3. DmMIC10b can polymerize into filamentous structures.**
**(A, B, C)** S2 cells expressing DmMIC10b-FLAG were immunolabeled against the FLAG epitope and analyzed by confocal microscopy and 2D STED nanoscopy. **(A)** Confocal recordings of cells with different expression levels of DmMIC10b-FLAG, increasing from left to right. The signal intensity reflects the expression level. **(B)** STED images of the cells shown in ((A), upper) and insets marked by dashed boxes ((A), lower). **(C)** Dual-color STED recordings of S2 cells labeled for DmMIC10b-FLAG (magenta) and DmMIC60 (green). Insets show magnified views of the areas indicated by dashed boxes. Scale bars: 2 μm.

As all DmMIC10b filaments shown thus far relied on the over-expression of a C-terminally FLAG-tagged version of DmMIC10b, we next expressed non-tagged DmMIC10b in mammalian HeLa, U-2 OS, and COS-7 cells and labeled them with antibodies against DmMIC10b to test whether the formation of the filamentous structures was induced by the FLAG epitope. Also, untagged DmMIC10b formed filaments at high expression levels, demonstrating that its propensity to form a filamentous structure is independent of the tag (Fig S6C–E).

We next investigated whether purified DmMIC10b is able to polymerize into filamentous structures in vitro. To this end, His$_6$-DmMIC10b was expressed in *Escherichia coli* and purified to homogeneity from inclusion bodies as described previously (Barbot et al, 2015). After the removal of detergent by dialysis, purified His$_6$-DmMIC10b exhibited a distinct ladder pattern on SDS–PAGE (Fig 4D, right panel), as previously reported for ScMIC10 (Barbot et al, 2015; Bohnert et al, 2015). This suggests that the purified DmMIC10b can associate into oligomers. In line with this observation, negative stain electron microscopy demonstrated the formation of filamentous structures upon the removal of the detergent (Fig 4E).

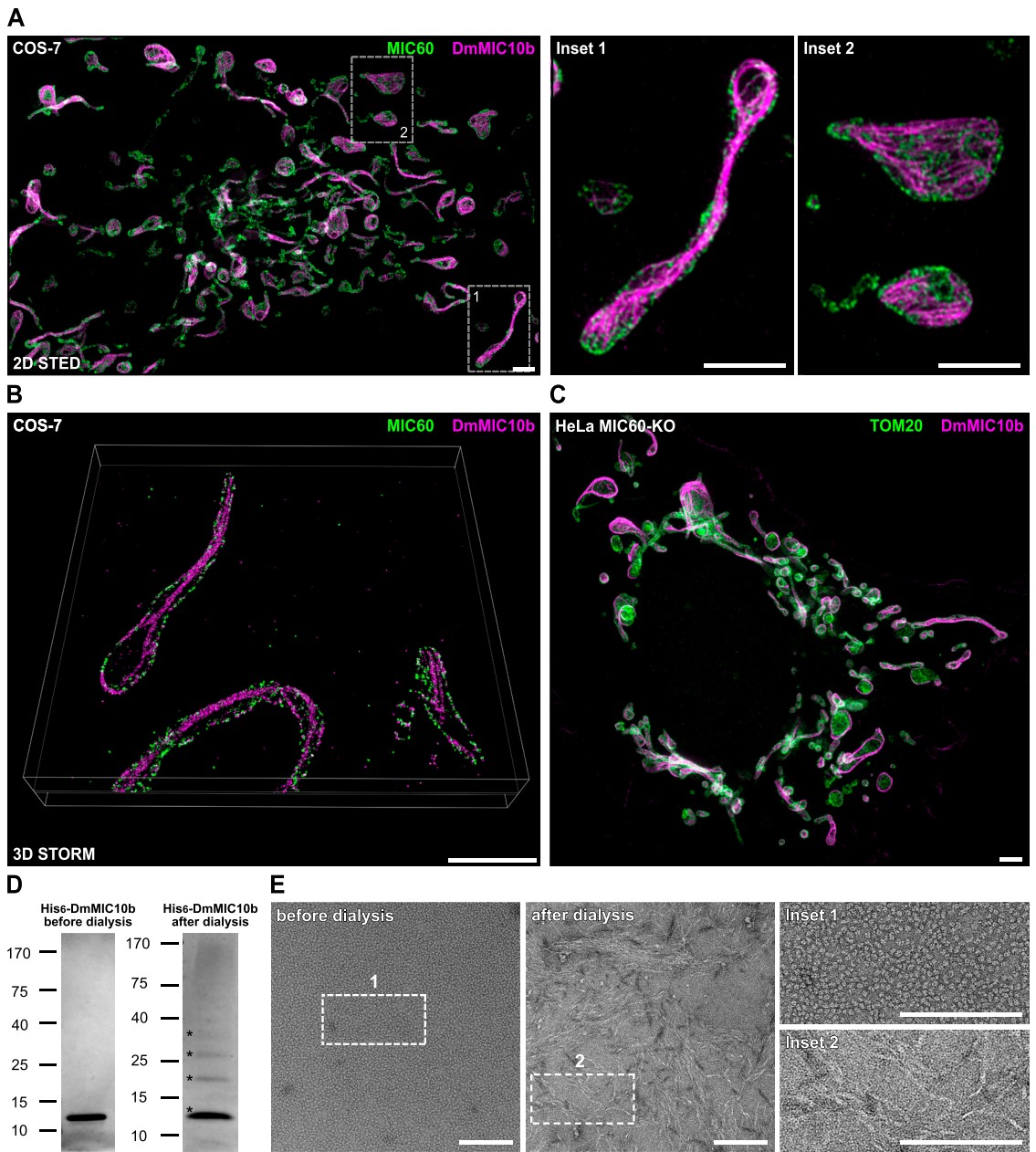

**Figure 4. DmMIC10b can form filaments in cells and in vitro.**
**(A, B, C)** Dual-color super-resolution microscopy of COS-7 cells expressing DmMIC10b-FLAG. Cells were fixed and immunolabeled with specific antibodies against the FLAG epitope (magenta) and MIC60 (green). **(A)** Representative 2D STED nanoscopy recording. Data were deconvolved. **(B)** Volume rendering of a 4Pi-STORM recording. Shown is a ~70-nm-thick cross section. **(C)** Representative dual-color STED recording of a MIC60-KO cell expressing DmMIC10b-FLAG. **(D, E)** His$_6$-DmMIC10b was purified from *E. coli* in the presence of urea, precipitated, and solubilized using sarcosyl. The detergent was removed by dialysis. **(D)** SDS–PAGE stained with Coomassie brilliant blue. Asterisks indicate different oligomeric species. **(E)** Same sample analyzed by negative stain transmission electron microscopy. Scale bars: 2 $\mu$m (A, B, C) and 150 nm (E).

Taken together, we conclude that DmMIC10b can polymerize into homo-oligomeric filaments both in mitochondria and in vitro.

## Filament formation seems to be specific for DmMIC10b

Although the formation of filamentous MICOS structures had been suggested by early studies conducted on yeast (Hoppins et al, 2011), several studies using super-resolution microscopy have not substantiated the existence of extended MICOS-only filaments in yeast or mammalian cells (Jakobs et al, 2020). Still, elongated MIC60 assemblies in mitochondria of COS-7 cells (Bates et al, 2022) or in mitochondria of HeLa cells depleted of the dynamin-like GTPase optic atrophy 1 (Stephan et al, 2020) have been reported. However, these ring- or arc-like structures only wrapped around the mitochondrial tubules and were by orders of magnitude shorter than the DmMIC10b filaments reported here. Previous studies overexpressing HsMIC10 did

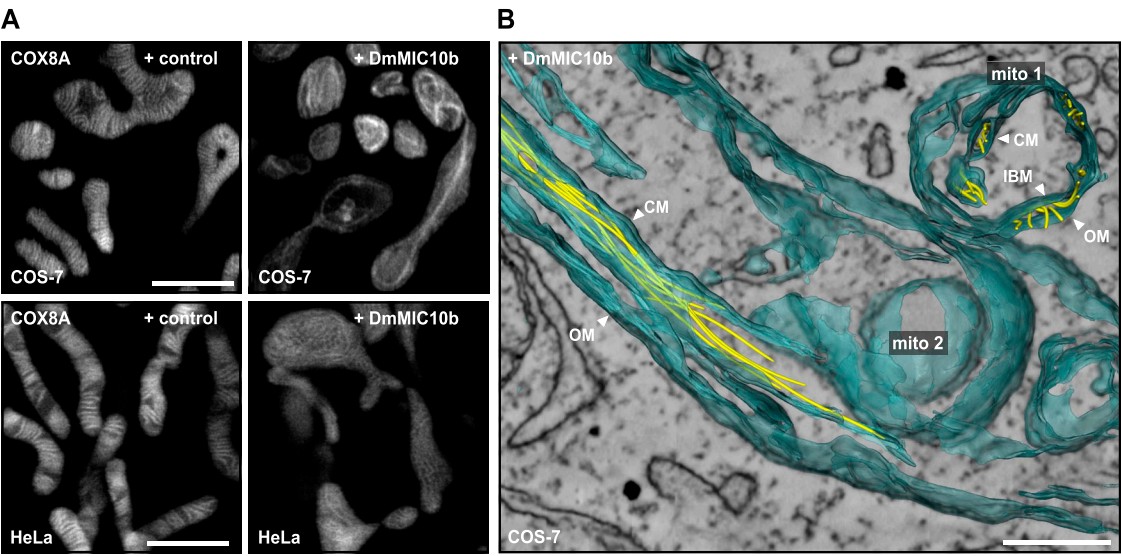

**Figure 5. DmMIC10b polymerizes into filaments, which alter the mitochondrial ultrastructure.**
**(A)** Live-cell STED recordings of mammalian cells expressing DmMIC10b. COS-7 and HeLa cells were co-transfected with DmMIC10b-FLAG and COX8A-SNAP to visualize the crista membranes. Cells were labeled with SNAP-cell SiR-647 and visualized by 2D live-cell STED nanoscopy. **(B)** Electron tomography of mitochondria from COS-7 cells expressing DmMIC10b-FLAG. The recording shows a cross section (mito-1) and a longitudinal section (mito-2) of two adjacent mitochondria. Membranes (blue) and filaments (yellow) were semi-automatically segmented and are shown as volume renderings. Abbreviations: OM, outer membrane; IBM, inner boundary membranes; CM, crista membrane. Scale bars: 2 μm (A) and 0.25 μm (B).

not report on the formation of filaments at various expression levels (Stephan et al, 2020) either. To further explore whether MIC10 from yeast or humans has a tendency to polymerize when expressed in a heterologous system, we expressed ScMIC10 in COS-7 cells and S2 cells and HsMIC10 in S2 cells. No filaments were observed, suggesting that the propensity to form filaments is a specific characteristic of DmMIC10b (Fig S7A–C).

## DmMIC10b filaments remodel the crista membranes

As MIC10 is a membrane-shaping protein (Barbot et al, 2015; Bohnert et al, 2015), it appeared possible that the DmMIC10b filaments influence the overall IM architecture. To test this, we expressed the cristae marker COX8A fused to a SNAP-tag (Stephan et al, 2019) together with DmMIC10b in HeLa and COS-7 cells and visualized the fusion protein inside mitochondria using live-cell STED nanoscopy. Both mitochondria of WT HeLa and COS-7 cells showed lamellar cristae as previously reported (Stephan et al, 2019; Liu et al, 2022). The overexpression of DmMIC10b strongly altered the cristae architecture, with the IM often apparently collapsed along the DmMIC10b filaments (Fig 5A).

To investigate the influence of DmMIC10b polymerization on the cristae architecture in more detail, we next recorded electron tomograms of chemically fixed COS-7 (Figs 5B and S8A) and S2 (Fig S8B) cells expressing DmMIC10b. Electron microscopy revealed filament bundles oriented along the mitochondria. As these filaments were absent in WT cells, we assume that these are DmMIC10b filaments. Whereas some of these filaments were in close contact with the IM, other filaments seemed to form between OM and IBM and inside the crista lumen, thereby widening the IMS or causing the formation of aberrant, tubular cristae (Figs 5B and S8). As STED

nanoscopy recordings of human MICOS knockout cells expressing DmMIC10b had indicated a similar arrangement of the DmMIC10b filaments (Fig S6A and B), we analyzed those cells using electron microscopy as well (Fig S9). In MIC60-KO cells, we observed most of the filaments in bundles between OM and IBM (Fig S9A). Despite its ability to interact with human MICOS, the expression of DmMIC10b could not rescue the aberrant cristae morphology in human MIC10-KO cells (Fig S9A–C). Instead, we observed the formation of bundles of filaments that remodeled the crista membranes in a similar way as observed in WT cells (Fig S9D).

Together, our data suggest that upon overexpression, DmMIC10b polymerizes into extended filaments that associate with bundles within the IMS and strongly influence the mitochondrial IM architecture.

## Filament formation of MIC10b is regulated by conserved amino acids

### Conserved glycine-rich motifs are a prerequisite for filament formation

Filament formation seemed to be a peculiarity of DmMIC10b; hence, we compared its primary sequence with that of MIC10 in humans, rats, zebrafish, and yeast (Fig 6A). MIC10 proteins exhibit highly conserved glycine-rich motifs in each of the two TM segments that mediate oligomerization in yeast (Barbot et al, 2015; Bohnert et al, 2015). Specifically, the first TMD of the investigated MIC10s contains a conserved GxxxG motif that is N-terminally extended into a GxGxGxG motif in humans, fish, and rats. In DmMIC10b, additional glycine residues extend this GxxxG motif into the sequence GxGGxxxG (GCGGVIIG). The second TMD contains a GxGxGxG motif that is conserved in all

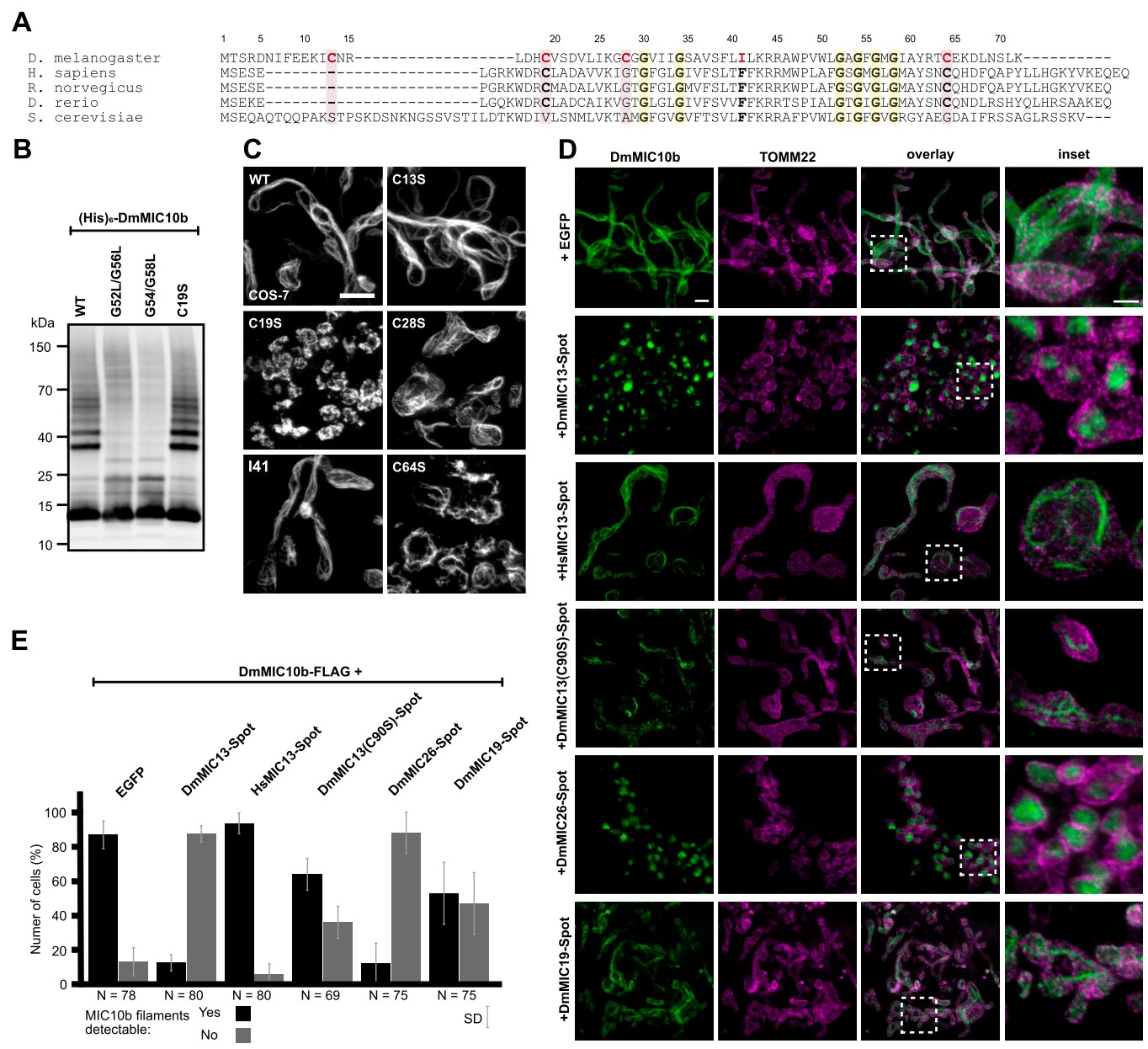

**Figure 6. Formation of DmMIC10b filaments depends on conserved amino acids and is suppressed by DmMICOS proteins.**
**(A)** Amino acid sequence alignment of MIC10 proteins from *D. melanogaster* and MIC10 from humans, rats, zebrafish, and yeast. **(B, C)** Amino acid residues exchanged for analysis in (B, C) are marked in red. **(B)** Western blot analysis of DmMIC10b mutants. His$_6$-DmMIC10b with point mutations was expressed in *E. coli*. Cells were homogenized, and the insoluble pellet was solubilized using sarcosyl. The detergent was removed by dialysis, and the samples were analyzed by SDS–PAGE and immunoblotting. **(C)** 2D STED nanoscopy of COS-7 cells expressing mutants of DmMIC10b-FLAG. Cells were chemically fixed and immunolabeled for the FLAG epitope. **(D, E)** Co-expression of MICOS proteins in COS-7 cells. Cells were co-transfected to induce the expression of DmMIC10b-FLAG together with other Spot-tagged MICOS proteins. Cells were immunolabeled against FLAG, Spot, and TOMM22. Double-transfected cells were recorded by dual-color STED nanoscopy. **(D)** Representative STED nanoscopy recordings. **(E)** Quantification of the number of cells, which formed DmMIC10b-FLAG filaments. Bars indicate the mean of three independent biological repeats, and whiskers indicate the SD. N indicates the total number of analyzed cells. Scale bars: 2 μm (C), 1 μm ((D), overview), and 0.5 μm ((D), inset).

investigated MIC10 proteins (Fig 6A). The well-investigated ScMIC10 from *S. cerevisiae* somewhat stands out as it exhibits an additional amino acid sequence of ~20 residues at its N-terminus (Fig 6A). Furthermore, all analyzed MIC10 proteins except ScMIC10 feature two highly conserved cysteine residues: a conserved cysteine in a distance of 11 aa N-terminally of the GxxxG motif (C19 in DmMIC10b) and a conserved cysteine that is

placed 6 aa C-terminally of the GxGxGxG motif (C64 in DmMIC10b). In addition, DmMIC10b uniquely features a cysteine residue (C28), which is part of the GxGGxxxG motif (GCGGVIIG) in the first TMD (Fig 6A). DmMIC10b also deviates from the other MIC10 proteins in a conserved sequence placed between the two predicted TMDs. Whereas all MIC10 proteins exhibit an FFKRR motif, phenylalanine 41 is replaced by isoleucine in DmMIC10b (Fig 6A).

As the glycine-rich motifs facilitate MIC10 oligomer formation in yeast (Barbot et al, 2015; Bohnert et al, 2015), we explored whether disruption of these motifs has any influence on the formation of the DmMIC10b filaments. To this end, mutations to disturb these motifs were introduced. Indeed, replacing the entire first glycine-rich motif in DmMIC10b (GCGGVIIG) with the human motif (IGTGFGLG) prevented the formation of filaments in DmMIC10b, but this mutation also strongly reduced the expression level of the protein (Fig S10A and B). Double-point mutations affecting one (G27L/G30L) or both (G30L/G34L) of the glycine residues of the core GxxxG motif prevented the formation of DmMIC10b filaments. However, both mutations interfered also with mitochondrial targeting, leading to the formation of some DmMIC10b structures outside of mitochondria (Fig S10A and B). When we introduced a single-point mutation outside the core GxxxG motif (G29L), this influenced neither mitochondrial targeting nor the formation of filaments (Fig S10A and B), suggesting that the core GxxxG motif in the first TMD of DmMIC10b is involved in mitochondrial targeting and possibly in filament formation.

Next, we analyzed the role of the highly conserved GxGxGxG motif placed in the second TMD of DmMIC10b. Concretely, we expressed DmMIC10b(G52L/G56L) and DmMIC10b(G54L/G58L) in S2 cells and in *E. coli* and analyzed their ability to form higher molecular weight species or filaments. After the expression of the proteins in *E. coli* cells, we lysed the cells, enriched inclusion bodies, and solubilized them with sarcosyl. After dialysis, we analyzed the samples using SDS–PAGE and Western blotting (Fig 6B). As previously reported for yeast MIC10, immunoblotting confirmed that disruption of the GxGxGxG motif in DmMIC10b strongly reduced the amount of higher molecular weight species preserved during SDS–PAGE (Barbot et al, 2015; Bohnert et al, 2015), suggesting the ablation of DmMIC10b oligomers. Both mutations abolished also filament formation, and similar to the mutations in the GxxxG motif, they hampered the import of the proteins into mitochondria as demonstrated by STED nanoscopy recordings (Fig S10C). Together, our findings suggest that DmMIC10b can form stable oligomers and that the existence of the glycine-rich motifs in both TMDs is a prerequisite for DmMIC10b filament formation in cells.

### Filament formation but not oligomer formation requires a conserved cysteine residue

Because DmMIC10b differs significantly from the well-studied ScMIC10 with respect to residues C13, C19, C28, C64, and I41, we investigated whether also mutations at these sites affect the ability of DmMIC10b to polymerize into filaments. To this end, the cysteine residues were individually replaced by a serine residue and the isoleucine by a phenylalanine residue. The variants were expressed in COS-7 and HeLa cells, and their localization was recorded by STED nanoscopy (Figs 6C and S11A and B). Except for DmMIC10b(C19S), which resulted in a mitochondrial, clustered, non-filamentous protein distribution, the other tested DmMIC10b variants were all capable of filament formation (Fig 6C). Therefore, we tested whether C19 is involved in the formation of stable MIC10 oligomers observed on Western blots (Fig 6B). In strong contrast to variants mutated in the GxGxGxG motif, DmMIC10b(C19S) behaved virtually identical to

WT DmMIC10b upon SDS–PAGE analysis (Fig 6B), suggesting that C19 has no influence on stable oligomer formation. We conclude that DmMIC10b oligomerizes into stable higher molecular weight species through its glycine-rich motifs as previously reported for MIC10 from yeast. The DmMIC10b filament formation relies on the conserved cysteine residue C19, suggesting that this amino acid can mediate the assembly of DmMIC10b oligomers into filaments.

### DmMICOS subunits can suppress filament formation of DmMIC10b

The formation of extended DmMIC10b filaments in fly cells was only observed upon the overexpression of DmMIC10b, which is a condition that results in a shifted balance between DmMIC10b and the other MICOS proteins. We speculated that at physiological conditions, filament formation of DmMIC10b is suppressed by interacting MICOS proteins. To explore this, DmMIC10b-FLAG was co-expressed with several Spot-tagged versions of MICOS proteins of the MIC10 and the MIC60 subcomplexes, and filament formation was determined by STED nanoscopy (Fig 6D). To ensure the balanced expression of the respective two overexpressed proteins, we expressed both proteins as a translational fusion, separated by a self-cleaving 2A peptide.

The co-expression of DmMIC10b with the MIC10 subcomplex subunits DmMIC13 or DmMIC26 suppressed DmMIC10b filament formation in most (>80%) of the cells, whereas DmMIC19, part of the MIC60 subcomplex, reduced filament formation less efficiently (~50%). Upon the co-expression of the human DmMIC13 orthologue HsMIC13, most of the analyzed cells still formed filaments, suggesting that the fly homologs have co-evolved to efficiently suppress the ability of DmMIC10b to form long filaments at physiological conditions (Fig 6D and E). The suppression of filaments by DmMIC13 was reduced when its only cysteine residue (C90) was replaced by a serine residue (Fig 6D and E), further supporting the notion that in *D. melanogaster*, cysteine residues are key for the regulation of the oligomerization status of MIC10.

Taken together, DmMIC10b, the major MIC10 from *D. melanogaster*, has the propensity to form extended filaments in vitro and in cells. Upon overexpression, the filaments reside in the IMS and deform the IM. The formation of filaments requires glycine-rich motifs within the TMDs, as well as a conserved cysteine residue (C19) in close proximity to the N-terminal TMD. The formation of filaments requires excess of DmMIC10b and is effectively suppressed by the co-overexpression of other constituents of the MIC10 subcomplex.

## Discussion

In this study, we show that the mitochondrial protein Dmel_CG41128/DmMIC10b is the major MIC10 orthologue in *D. melanogaster*. DmMIC10b is ubiquitously expressed and interacts with the MICOS complexes of flies, humans, and monkeys. It controls the stability of the MIC10 subcomplex of *D. melanogaster* and is required for maintaining the mitochondrial ultrastructure. Despite its high homology to ScMIC10 and HsMIC10, DmMIC10b stands out because of its propensity to polymerize into extended

filaments. At low expression levels, DmMIC10b formed distinct clusters, resembling the distribution of MICOS proteins in yeast or human cells (Jans et al, 2013; Stoldt et al, 2019; Kondadi et al, 2020; Pape et al, 2020; Bates et al, 2022), whereas at high expression levels, DmMIC10b polymerized into bundles of filaments, which influenced both the fusion–fission balance of the mitochondrial network and the cristae architecture. Remarkably, these filaments were located between the OM and IM, as well as inside the crista lumen, suggesting that they can form or extend outside of the contiguous IM.

Filaments formed by DmMIC10b resemble those formed by the GFP-tagged bacterial cytoskeleton protein Mreb (Grotjohann et al, 2011; Pande et al, 2022). The formation of extended Mreb filaments has been described as an artifact caused by tagging with fluorescent proteins (Swulius & Jensen, 2012). However, at physiological conditions, untagged Mreb polymerizes into smaller arc-like or ring-like assemblies, which are crucial to determine the bacterial cell shape (Shi et al, 2018). This study shows that the formation of DmMIC10b filaments is independent of the C-terminal tag. Instead, polymerization into filaments depends primarily on the expression level and thereby concentration of DmMIC10b.

Similar to yeast MIC10 (Bohnert et al, 2015), DmMIC10b relies on conserved glycine-rich motifs to form oligomeric species that are preserved during SDS–PAGE analysis. In the case of DmMIC10b, this oligomerization seems to be required also for the assembly of extended filaments. Importantly, we found that a highly conserved cysteine residue (C19 in DmMIC10b), located N-terminally of the first TMD of DmMIC10b, is crucial for the formation of filaments, but not for the generation of stable oligomers whose formation is mediated by the glycine-rich motifs. Remarkably, this cysteine residue, as well as the cysteine residue located C-terminally of the second TM segment (C64 in DmMIC10b), is highly conserved in the MIC10 proteins of higher animals, including humans, flies, rats, and fish, whereas the well-studied MIC10 from yeast does not contain any cysteine residue (Fig 6A). This may point to understudied differences between the MICOS complexes of lower and higher eukaryotes. Given the sequence homology of the MIC10 proteins in higher eukaryotes, the question remains why DmMIC10b has the propensity to form filaments, whereas the human MIC10 seems not to have this ability. A crystal structure of MIC10 is not available, but models generated by AlphaFold2 may provide some hints (Fig 1C). Structure predictions suggest that DmMIC10b, HsMIC10, and ScMIC10 all feature a hairpin-like topology, but differences exist regarding the length and shape of the two α-helices that contain the TMDs and the conserved glycine-rich motifs. The predictions suggest that in DmMIC10b, the N-terminal α-helix, which contains the conserved C19 residue, is elongated and exhibits a curved shape compared with other MIC10 proteins. Moreover, DmMIC10b does not seem to feature the flexible termini predicted for human or yeast MIC10. We speculate that such differences in the shape of the MIC10 monomers influence the shape of MIC10 oligomers and their ability to polymerize into filaments.

We found that in fly cells, the stoichiometric ratio between DmMIC10b and other MICOS subunits influences the propensity of DmMIC10b to polymerize into extended filaments. This seems to be an effective mechanism to keep MIC10 oligomerization in check. Indeed, we never observed extended DmMIC10b filaments in WT fly cells, which does not exclude the possibility that at specific developmental stages, the ratio between DmMIC10b and the other MICOS subunits is changed and filaments are formed to rearrange the mitochondrial architecture. Therefore, it will be revealing to investigate the interplay of cristae-shaping proteins along all developmental stages of an organism such as a fly.

# Materials and Methods

## Plasmids

### Plasmids for generation of DmMIC10b knockout

gRNA sequences upstream of the CG41128 5′-UTR and downstream of the CG41128 3′-UTR were identified using the web-based tool introduced by Gratz et al (2014) (https://flycrspr.org/target-finder/).

Criteria were as follows: CRISPR targets with 5′-G; Stringency: High; PAM: NGG Only and the closest site, with no off-target to the UTR. The respective sequences were purchased as 5′-phosphorylated oligonucleotides, annealed, and ligated into the BbsI sites of pU6-BbsIchiRNA (45946; Addgene) (Gratz et al, 2014) to create pU6-BbsI-chiRNA-CG41128-5′gRNA (oligonucleotides 1/2, Table 1) and pU6-BbsI-chiRNA-CG41128-3′gRNA (oligonucleotides 3/4, Table 1).

The donor plasmid pHD-attP-DsRed-CG41128 for homologous recombination was generated using Gibson assembly. To this end, pHD-DsRed-attP (51019; Addgene) (Gratz et al, 2014) was double-digested with NotI/XhoI and used as the vector backbone. Genomic DNA of a WT strain (w⁻) was used as a template to amplify homology arms of about 1 kbp flanking the cleavage sites, and the missing base pairs from the cleavage site to the corresponding UTR beginning/end. The DsRed-attP fragment was amplified from pHD-DsRed-attP. The NEBuilder Assembly Tool was used to design the oligonucleotides and Gibson Assembly Master Mix (NEB) for the assembly reaction.

The 5′-homology arm was amplified using the oligonucleotides 5/6 (Table 1). The 3′-homology arm was amplified using the oligonucleotides 13/14 (Table 1). The 5′-missing arm was amplified using the oligonucleotides 7/8 (Table 1). The 3′-missing arm was amplified using the oligonucleotides 11/12 (Table 1). The DsRed-attP fragment was amplified using the oligonucleotides 9/10 (Table 1).

## Expression plasmids for S2 cells

### pUAS-CG41128

DmMIC10b was amplified from cDNA using the oligonucleotides 15/16 (Table 1) and ligated into the EagI/XhoI restrictions sites of pUASPattB (a gift from Alf Herzig, MPI for Biophysical Chemistry, Goettingen).

### pUAS-CG41128-FLAG

DmMIC10b-FLAG was amplified from cDNA using the oligonucleotides 15/17 (Table 1) and ligated into the EagI/XhoI restriction sites of pUASPattB.

**Table 1.  List of oligonucleotides for PCR.**

| Oligo | Oligo name | Sequence (5′–3′) | Internal |
|---|---|---|---|
| 1 | 28-5-Fw | CTTCGAAGCCAGTTTGCAAAAGGA | 7669 |
| 2 | 28-5-Rev | AAACTCCTTTTGCAAACTGGCTTC | 7670 |
| 3 | 28-3-Fw | CTTCGACTATTCGTTGTTAGTTTA | 7671 |
| 4 | 28-3-Rev | AAACTAAACTAACAACGAATAGTC | 7672 |
| 5 | 28-5′-arm Fw | GCAGGTGGAATTCTTGCATGCTAGCAAACATGATATAAGAGACCG | 7651 |
| 6 | 28-5′-arm Rev | CTCCTGTCATTCCTTTTGCAAACTGGCTTCTAC | 7652 |
| 7 | 28-5′-missing part Fw | CCAGTTTGCAAAAGGAATGACAGGAGGGCATGG | 7653 |
| 8 | 28-5′-missing part Rev | GGGCACTACGATCCTTTGTAAATTTCGATGTGCGTCAAG | 7654 |
| 9 | 28-attP-DsRed Fw | GAAATTTACAAAGGATCGTAGTGCCCCAACTGG | 7655 |
| 10 | 28-attP-DsRed Rev | AAATAAGTATATTTATAACTTCGTATAGCATACATTATA CGAAGTTATACC | 7656 |
| 11 | 28-3′-missing arm Fw | ATACGAAGTTATAAATATACTTATTTAGTGCT TATTAATACTG | 7657 |
| 12 | 28-3′-missing arm Rev | ATTCGTTGTTAGTTTACAGGTTTATGGGTGATTTTTTTC | 7658 |
| 13 | 28-3′-arm Fw | CATAAACCTGTAAACTAACAACGAATAGTCAAAATG | 7659 |
| 14 | 28-3′-arm Rev | CTTGAACTCGATTGACGGAAGA GCCTTACAAAGGATGGACTGAGAAC | 7660 |
| 15 | EagI-DmMIC10b Fw | TTCGGCCGATGACTTCTCGCGATAATATTTTCG | 7794 |
| 16 | DmMIC10b-XhoI Rev | TTTACTCGAGTCATTTTAAAGAATTTAAATCCTTTTC | 8099 |
| 17 | DmMIC10b-FLAG-XhoI Rev | TTTACTCGAGTCACTTGTCGTCATCGTCTTTGTAG TCACCAGAGCCTCCTTTTAAAGAATTTAAATCCTTTTC | 7632 |
| 18 | Mic10b-G52/56L Fw | GCCTGTATGGCTCCTCGCTGGATTTTTAATGGGCATCGCTTATAG | 9580 |
| 19 | Mic10b-G52/56L Rev | CTATAAGCGATGCCCATTAAAAATCCAGCGAGGAGCCATACAGGC | 9581 |
| 20 | Mic10b-G54/58L Fw | GGCTCGGCGCTCTATTTGGAATGCTCATCGCTTATAGG | 9582 |
| 21 | Mic10b-G54/58L Rev | CCTATAAGCGATGAGCATTCCAAATAGAGCGCCGAGCC | 9583 |
| 22 | EagI-hMic10 Fw | TTCGGCCGATGTCTGAGTCGGAGCTC | 8237 |
| 23 | HMic10-FLAG Rev | TTTACTCGAGTCACTTGTCGTCATCGTCTTTGTAG TCACCAGAGCCTCCCTGCTCCTGCTCTTTGAC | 8226 |
| 24 | EagI-YMic10 Fw | TTCGGCCGATGTCCGAACAAGCACAAACAC | 8238 |
| 25 | Xho-YMic10-FLAG Rev | TTTACTCGAGTCACTTGTCGTCATCGTCTTTGTAG TCACCAGAGCCTCCAACCTTCGAGGATCTGAGGC | 8239 |
| 26 | DmMIC10b Fw | ATAAAGCTTGCCGCCACCATGACATCT | 10084 |
| 27 | DmMIC10b-FLAG Rev | AAACTCGAGTTACTTATCGTCATCGTCCTTATA | 10085 |
| 28 | DmMIC10b(C13S) Fw | GGAGAAGATCTCCAATCGCCTGG | 10175 |
| 29 | DmMIC10b(C13S) Rev | TCGAAGATGTTGTCCCTAG | 10176 |
| 30 | DmMIC10b(C19S) Fw | CCTGGACCACTCCGTGAGCGATG | 10177 |
| 31 | DmMIC10b(C19S) Rev | CGATTGCAGATCTTCTCCTCGAAG | 10178 |
| 32 | DmMIC10b(C28S) Fw | GATCAAGGGATCCGGAGGCGTGA | 10179 |
| 33 | DmMIC10b(C28S) Rev | AGCACATCGCTCACGCAG | 10180 |
| 34 | DmMIC10b(I41F) Fw | GTCCTTCCTGTTCCTGAAGAGGAG | 10183 |
| 35 | DmMIC10b(I41F) Rev | ACGGCGCTTCCGATGATC | 10184 |
| 36 | DmMIC10b(C64S) Fw | CTACCGGACCTCCGAGAAGGACC | 10181 |
| 37 | DmMIC10b(C64S) Rev | GCGATTCCCATTCCAAATC | 10182 |
| 38 | DmMIC10b(G29L) Fw | CAAGGGATGCCTGGGCGTGATCATC | 11737 |
| 39 | DmMIC10b(G29L) Rev | ATCAGCACATCGCTC | 11738 |

**Table 1. Continued**

| Oligo | Oligo name | Sequence (5′–3′) | Internal |
|---|---|---|---|
| 40 | DmMIC10b(G27/30L) Fw | GGACTGGTGATCATCGGAAGC | 11739 |
| 41 | DmMIC10b(G27/30L) Rev | GCACAGCTTGATCAGCACATCG | 11740 |
| 42 | DmMIC10b(G30/34L) Fw | TCATCCTGAGCGCCGTGTCCTTC | 11741 |
| 43 | DmMIC10b(G30/34L) Rev | TCACCAGTCCGCATCCCTTGATC | 11742 |
| 44 | DmMIC10b(hum. GX1) Fw | GTTTTGGATTAGGAAGCGCCGTGTC | 11743 |
| 45 | DmMIC10b(hum. GX1) Rev | CAGTACCTATCTTGATCAGCACATCGC | 11744 |
| 46 | TRE3G-DmMIC10b Fw | GCGCAAACTAGTATGTGTTCGATTCTAGATTCGAG | 9965 |
| 47 | TRE3G-DmMIC10b Rev | GCGCAAACCGGTCTTCAGGCTGTTCAGGTCCTTCTC | 9966 |
| 48 | EagI-ScMic10 Fw | GCGCAACGGCCGCCATGTCCGAACAAGCACAAACAC | 10068 |
| 49 | ScMic10-FLAG-XbaI Rev | GCGCAATCTAGATCACTACTTGTCGTCATCGTCTT TGTAGTCGCCTCCAACCTTCGAGGATCTGAGGCCAGCG | 10069 |
| 50 | Mic13_C90S Fw | CATGCTGCCCTCCTACGCAGGCA | 10250 |
| 51 | Mic13_C90S Rev | TGGATGAAACGGAAGGTGTTCTTCAC | 10251 |
| 52 | SalI-DmMic13 Fw | GCGCAAGTCGACATGGTTCTAGGATTTCTAGTGCGCG | 9921 |
| 53 | DmMic13-MluI Rev | GCGCAAACGCGTTTAGCTGCTCCAATGGCTCACGGCG | 9922 |
| 54 | SalI-HsMic13 Fw | GCGCAAGTCGACATGGTGGCCCGGGTGTGGTCGC | 10098 |
| 55 | HsMic13-MluI Rev | GCGCAAACGCGTCTAGCTGCTCCAATGGCTCACG GCGCGCACGCGGTCTGGCTTGGTGCGCGCCTTCACATACTCCC | 10099 |
| 56 | SalI-DmMic26 Fw | GCGCAAGTCGACATGCTGCGCAAAACGGCAACGATGG | 9925 |
| 57 | DmMic26-MluI Rev | GCGCAAACGCGTTTAGCTGCTCCAATGGCTCACGGCG | 9926 |
| 58 | SalI-DmMic19 Fw | GCGCAAGTCGACATGGGAGCCCGACAGTCTCAATCCC | 9923 |
| 59 | DmMic19-MluI Rev | GCGCAAACGCGTCTAGCTGCTCCAATGGCTCACGGCG | 9924 |
| 60 | CG41128 Fw | AAAGGCCATGGAAATGACTTCTCGCGATAATAT | 6337 |
| 61 | CG41128 Rev | GCGAATTCAAATTATTTTAAAGAATTTAAATCCTTTTCAC | 6338 |

### pUAS-CG41128(G52L/G56L)-FLAG and pUAS-CG41128(G54L/G58L)-FLAG

Glycine mutants were produced by site-directed mutagenesis of pUAS-CG41128-FLAG using the oligonucleotides 18/19 or 20/21 (Table 1).

### pUAS-HsMIC10-FLAG

HsMIC10-FLAG was amplified by PCR using the oligonucleotides 22/23 (Table 1) and ligated into the EagI and XhoI restriction sites of pUASPattB.

### pUAS-ScMIC10-FLAG

ScMIC10-FLAG was amplified by PCR using the oligonucleotides 24/25 (Table 1) and integrated into the EagI and XhoI restriction sites of pUASPattB.

### Expression plasmids for mammalian cells

### pcDNA3.1-DmMIC10b-FLAG

Humanized DmMIC10b-FLAG (GenScript) was amplified by PCR using the nucleotides 26/27 (Table 1) and inserted into the HindIII and XhoI restriction sites of pcDNA3.1(+) (Thermo Fisher Scientific). The point mutants MIC10b(C13S)-FLAG, DmMIC10b(C19S)-FLAG, DmMIC10b(C28S)-FLAG, DmMIC10b(I41F)-FLAG, DmMIC10b(C64S)-FLAG, DmMIC10b(G29L)-FLAG,

DmMIC10b(G27L/G30L)-FLAG, DmMIC10b(G30L/G34L)-FLAG, and DmMIC10b(hum. GX1)-FLAG (humanized first TM segment) were produced using the site-directed mutagenesis kit (NEB) and the oligonucleotides 26–45 (Table 1).

### AAVS1-TRE3G-DmMIC10-FLAG-T2A-EGFP

Humanized DmMIC10b-FLAG flanked by AgeI and EcoRV restriction sites (obtained from GenScript) was inserted into the AgeI/EcoRV restriction sites of AAVS1-TRE3G-MIC10-FLAG-T2A-EGFP (Stephan et al, 2020).

### AAVS1-TRE3G-DmMIC10-T2A-EGFP

TRE3G-DmMIC10 was amplified (including SpeI and AgeI restriction sites) from AAVS1-TRE3G-DmMIC10b-FLAG-T2A-EGFP by PCR using the oligonucleotides 46/47 (Table 1). AAVS1-TRE3G-MIC10-FLAG-T2A-EGFP (Stephan et al, 2020) was linearized using SpeI and AgeI restriction endonucleases, and the PCR product was ligated into the plasmid to remove the FLAG epitope.

### pFLAG-CMV5.1-ScMIC10-FLAG

ScMIC10-FLAG was amplified by PCR using the oligonucleotides 48/49 (Table 1) and integrated into the EagI/XbaI restriction sites of pFLAG-CMV5.1 (Sigma-Aldrich).

## Co-expression of DmMIC10b-FLAG and Spot-tagged MICOS proteins

### DmMIC13-(C90S)-Spot_pJET1.2

The substitution mutation DmMIC13(C90S)-Spot was introduced to DmMIC13-Spot by site-directed PCR mutagenesis. To this end, DmMIC13-Spot (GenScript) was amplified by PCR using the oligonucleotides 50/51 (Table 1) to introduce the substitution mutation C90S into DmMIC13-Spot.

### AAVS1-TRE3G-DmMIC10b-FLAG-T2A-DmMIC13-Spot and AAVS1-TRE3G-DmMIC10b-FLAG-T2A-DmMIC13(C90S)-Spot

DmMIC13-Spot or DmMIC13(C90S)-Spot was amplified by PCR using the oligonucleotides 52/53 (Table 1). AAVS1-TRE3G-DmMIC10b-FLAG-T2A-EGFP was linearized using SalI/MluI restriction endonucleases, and DmMIC13-Spot or DmMIC13(C90S)-Spot was ligated into the backbone.

### AAVS1-TRE3G-DmMIC10b-FLAG-T2A-HsMIC13-Spot

HsMIC13 was amplified by PCR using the oligonucleotides 54/55 (Table 1). AAVS1-TRE3G-DmMIC10b-FLAG-T2A-EGFP was linearized using SalI/MluI restriction endonucleases, and HsMIC13-Spot was ligated into the backbone.

### AAVS1-TRE3G-DmMIC10b-FLAG-T2A-DmMIC26-Spot

DmMIC26-Spot (GenScript) was amplified by PCR using the oligonucleotides 56/57 (Table 1). AAVS1-TRE3G-DmMIC10b-FLAG-T2A-EGFP was linearized using SalI/MluI restriction endonucleases, and DmMIC26-Spot was ligated into the backbone.

### AAVS1-TRE3G-DmMIC10b-FLAG-T2A-DmMIC19-Spot

DmMIC19-Spot (GenScript) was amplified by PCR using the oligonucleotides 58/59 (Table 1). AAVS1-TRE3G-DmMIC10b-FLAG-T2A-EGFP was linearized using SalI/MluI restriction endonucleases, and DmMIC19-Spot was ligated into the backbone.

## Expression plasmids for bacteria

### pPROEXHTb-DmMIC10b and derived point mutants

DmMIC10b cDNA was amplified by PCR using oligonucleotides 60/61 (Table 1) and integrated into pPROEXHTb (Thermo Fisher Scientific) via the NcoI/EcoRI restriction sites. Point mutants were generated by site-directed mutagenesis PCR.

## Purification of DmMIC10b from *E. coli* and preparation for EM

The purification of His-tagged DmMIC10b was performed as described previously (Barbot et al, 2015). In brief, *E. coli* BL21 (DE3) cells were collected by centrifugation after the expression of His$_6$-DmMIC10b (1 mM isopropyl-$\beta$-D-thiogalactopyranoside [IPTG], 3 h, 37°C) and stored at –20°C until purification. After thawing, cells were lysed, and inclusion bodies were isolated and subsequently dissolved in resuspension buffer containing 8 M urea, 150 mM NaCl, 20 mM Tris–HCl, 40 mM imidazole, and 0.5 mM DTT, pH 8.0. The mixture was applied to a HisTrap column (5 ml) and eluted with resuspension buffer supplemented with 500 mM imidazole. Isolated MIC10 was further subjected to a HiLoad 16/600 Superdex 200

size-exclusion column (GE Healthcare). Separated fractions were analyzed by SDS–PAGE and Coomassie brilliant blue staining.

## Expression of DmMIC10b mutants in *E. coli* and analysis by Western blotting

DmMIC10b mutants were expressed in *E. coli* BL21 (DE3). Cells were collected by centrifugation after expression (1 mM isopropyl-$\beta$-D-thiogalactopyranoside [IPTG], 3 h, 37°C) and washed in salt buffer containing 150 ml NaCl and 10 mM Hepes, pH 7.4. The pellet was resuspended in lysis buffer containing 50 mM Tris (pH 8.0), 150 mM NaCl, 0.1 mg/ml lysozyme, 1 mM MgCl$_2$, and cOmplete protease inhibitor (Merck), and the cells were homogenized by sonication. The homogenate was supplemented with Benzonase (Sigma-Aldrich) and stirred for 30 min at 4°C. After centrifugation, the pellet was washed with Triton X-100 wash buffer containing 50 mM Tris, 150 mM NaCl, 1 mM EDTA, and 2% (wt/vol) Triton X-100 (pH 8.0) followed by washing with a wash buffer containing 50 mM Tris, 100 mM NaCl, 10 mM DTT, and 1 mM EDTA (pH 8.0). The pellet was dissolved with 8% (wt/vol) sarcosyl and 1 M urea (in 50 mM Tris, pH 8.0), and insoluble material was removed by centrifugation. The solution was diluted with solubilization buffer (20 mM Tris and 150 mM NaCl, pH 8.0) to a final concentration of 1.5% (wt/vol) sarcosyl, supplemented with 0.1% (wt/vol) DDM, and dialyzed against solubilization buffer overnight. Samples were analyzed by SDS–PAGE followed by Western blotting. His-tagged DmMIC10b was detected using an anti-His antibody (34660; QIAGEN N.V.).

## Cultivation of flies and life span assay

Flies were maintained on standard fly food with cornmeal, yeast, and agar at 25°C on a 12/12-h light/dark cycle. Fly stocks used were *DmMIC10b[KO]* (this study), *w[1118]* (#6326; Bloomington Drosophila Stock Center [BDSC]), and OR-R (#5; BDSC). WT control flies used were the heterozygous progeny of *w[1118]* males and OR-R females.

The life span assay was performed using four biological replicates per genotype. 15 male and 15 female young flies were placed in each vial. Every 2–4 d, flies were transferred to new food vials, and both living and dead flies were counted for consistency. Counting was continued until all flies had died.

## Generation of knockout flies

*Dmel_CG41128[KO]* flies were generated by a commercial transformation service (BestGene Inc). The injection stock was RRID: BDSC_55821 (Bloomington Drosophila Stock Center).

## Cell culture

*Drosophila* S2 cells (Cat. No. R69007, Lot No. 2082623; Thermo Fisher Scientific/Gibco) were cultivated in Schneider's *Drosophila* medium (Cat. No. 21720024; Merck/Sigma-Aldrich/Gibco) supplemented with 1mM sodium pyruvate (Cat. No. S8636; Merck/Sigma-Aldrich) and 10% (vol/vol) fetal bovine serum (Cat. No. FBS. S 0615, FBS Superior stabil; Bio & Sell GmbH). Cells were cultivated at 28°C and ambient CO$_2$ levels. Kidney fibroblast-like cells (COS-7) from the green African monkey *C. aethiops* (Cat. No. 87021302, Lot No. 05G008; Merck/Sigma-

Aldrich) and human cervical cancer cells (HeLa) were cultivated in DMEM, containing 4.5 g/liter glucose and GlutaMAX additive (Cat. No. 10566016; Thermo Fisher Scientific) supplemented with 1 mM sodium pyruvate (Cat. No. S8636; Merck/Sigma-Aldrich) and 10% (vol/vol) FBS at 37°C and 5% $CO_2$. Human osteosarcoma cells (U-2 OS) (Cat. No. 92022711, ECACC) were cultivated at 37°C and 5% $CO_2$ and grown in McCoy's medium (Thermo Fisher Scientific) supplemented with 10% (vol/vol) FBS and 1% (vol/vol) sodium pyruvate.

### Transfection of HeLa, COS-7, and S2 cells

S2 cells were split the day before transfection. On the day of transfection, $1.5 \times 10^6$ cells were seeded per well of a six-well dish. Transient transfection was achieved using Effectene transfection reagent (Cat. No. 301425; QIAGEN) and 2 μg of plasmid DNA per well. After 23 h, the cells were detached and reseeded on glass coverslips coated with concanavalin A (Cat. No. C5275; Merck/Sigma-Aldrich) for 1 h.

Mammalian cells were seeded on coverslips (for light microscopy), on Aclar film (for electron microscopy), and in cell culture dishes (for biochemistry) 1 d before transfection. Transient transfection was achieved by jetPRIME transfection reagent (Cat. No. 114-15; Polyplus-transfection) following the manufacturer's protocol.

### Generation of antibodies against DmMICOS proteins

*Drosophila* anti-MICOS polyclonal antibodies were produced by injecting purified His-tagged DmMIC10b or synthetic peptides into rabbits (Prof. Hermann Ammer, Munich, Germany). For anti-MIC10 antibody generation, recombinantly expressed and affinity-purified MIC10 protein, and for anti-MIC60 and anti-MIC13 antibody generation, AAKPKDNPLPRDVVELEKA and GDSDQTDKLYNDIKSELRPH synthetic peptides were injected into rabbits, respectively. All antibodies were affinity-purified.

### Mitochondrial isolation

Mitochondria were isolated as described previously (Panov & Orynbayeva, 2013) with slight modifications. Flies were taken up in cold TH buffer, containing 300 mM trehalose, 10 mM KCl, 10 mM Hepes (pH 7.4), 2 mM PMSF, 0.1 mg BSA/ml, and protease inhibitor mix (04693116001; Roche). The flies were then two times homogenized with a Dounce homogenizer (800 rpm/min), and the large remaining fragments were pelleted at 400*g* for 10 min at 4°C. Afterward, the remaining pieces were removed by centrifugation (800*g*, 8 min, 4°C) and the mitochondria-containing supernatant was saved in a new vial. To collect mitochondria, a further spin at 11,000*g*, 10 min, 4°C was performed, and pellets from separate reaction tubes were pooled and washed with BSA-free TH buffer. The mitochondrion concentration was determined using the Bradford assay.

### Affinity purification from fly mitochondria

Isolated mitochondria (1 mg) were transferred to lysis buffer (150 mM NaCl, 10% glycerol [vol/vol], 20 mM $MgCl_2$, 2 mM PMSF,

50 mM Tris–HCl, pH 7.4, 1% digitonin [vol/wt], and protease inhibitor [04693116001; Roche]) and agitated for 30 min at 4°C. Debris was removed by centrifugation (15 min, 16,000*g*, 4°C), and the cleared supernatant was transferred on protein A-Sepharose conjugated with anti-DmMIC60 or control antisera. After 1 h binding at 4°C, beads were washed 10 times with washing buffer (50 mM Tris–HCl, pH 7.4, 150 mM NaCl, 10% glycerol [vol/vol], 20 mM $MgCl_2$, 1 mM PMSF, and 0.3% digitonin [wt/vol]). Proteins were eluted by the addition of 0.1 M glycine, pH 2.8, at RT for 5 min. Eluates were neutralized and analyzed by SDS–PAGE and immunoblotting using specific antibodies against DmMIC60 (this study), DmMIC10b (this study), DmMIC13 (this study), NDUFA9, and RIESKE (Dennerlein et al, 2021).

### Affinity purification from cell lysates

COS-7 and HeLa cells were seeded in 15-cm cell culture dishes and cultured overnight. Cells were transfected with AAVS1-TRE3G-MIC10-FLAG-T2A-EGFP (Stephan et al, 2020, EMBOJ), AAVS1-TRE3G-DmMIC10-FLAG-T2A-EGFP, or AAVS1-TRE3G-EGFP (Qian et al, 2014) (52343; Addgene). The medium was exchanged 4 h after transfection, and expression was induced by adding 1 μg/ml doxycycline hyclate for 24 h. Cells were harvested by trypsinization and centrifugation. All the following steps were performed at 4°C. The pellet was washed with PBS and resuspended in 1.5 ml lysis buffer containing 20 mM Tris (pH 7.0), 100 mM NaCl, 1 mM EDTA, 10% (vol/vol) glycerol, 1% (wt/vol) digitonin (Carl Roth GmbH), and cOmplete protease inhibitor (Merck). Samples were rotated on a wheel for 1 h. After centrifugation at 13,000*g* for 10 min, the supernatant was transferred onto equilibrated magnetic anti-FLAG M2 magnetic beads (Merck). Samples were rotated on a wheel for 105 min. The supernatant was removed, and beads were washed 10 times with wash buffer containing 20 mM Tris, 100 mM NaCl, 1 mM EDTA, 5% (vol/vol) glycerol, 0.25% (wt/vol) digitonin, and cOmplete protease inhibitor (pH 7.0). Elution was performed by adding 0.1 M glycine (pH 2.8) and shaking the samples at 1,200 rpm for 20 min. Eluates were neutralized and analyzed by SDS–PAGE and immunoblotting using specific antibodies against FLAG (F3165; Sigma-Aldrich), MIC13 (SAB1102836; Sigma-Aldrich), MIC19 (HPA042935; Atlas Antibodies), COX3 (55082-1-AP; Proteintech), MIC60 (10179-1-AP; Proteintech), and MFN1 (ab57602; Abcam).

### Preparation of fly tissues for fluorescence microscopy

Adult testes were dissected in ice-cold PBS, then fixed at RT using 4% formaldehyde in PBT (PBS + 0.2% [vol/vol] Triton X-100) for 20 min. After fixation, testes were rinsed in PBT (multiple rinses for a minimum of 30 total minutes), then blocked using PBTB (PBT + 0.2% [wt/vol] BSA + 5% [vol/vol] normal goat serum) for 30 min. After blocking, samples were incubated in primary antibody diluted in PBTB overnight at 4°C. Then, the primary antibody was removed, and samples were rinsed in PBT for 30 min, then for 30 min in PBTB. Secondary antibodies diluted 1:500 in PBTB were added to the samples overnight at 4°C. Samples were then rinsed again with PBT for 30 min, treated with DAPI for 10 min to stain DNA, and finally rinsed again with PBT. Samples were stored in Vectashield (Vector Laboratories) until being mounted on slides and imaged. Primary

antibodies used for staining testes were rabbit anti-DmMIC10b (1:3,000, this work) and mouse anti-ATP5A (1:300, ab14748; Abcam).

## Sample preparation of fixed S2 cells for STED nanoscopy

Fixation and labeling of S2 cells were done as described previously (Wurm et al, 2010). In brief, cells were fixed using an 8% (wt/vol) formaldehyde solution, permeabilized by incubation with a 0.25% (vol/vol) Triton X-100 solution, and blocked with a 5% (wt/vol) BSA solution. Proteins of interest were labeled with antisera against the FLAG-tag (Thermo Fisher Scientific) and ATP5A (Abcam). Detection was achieved via secondary antibodies custom-labeled with Alexa Fluor 594 or STAR RED.

## Sample preparation of fixed HeLa and COS-7 cells for STED nanoscopy and 4Pi-STORM

Cells were transfected with pcDNA3.1-DmMIC10b-FLAG or with AAVS1-TRE3G-DmMIC10b-FLAG-T2A-EGFP. In case of the latter construct, cells were induced with 1 μg/ml doxycycline. Fixation and labeling were performed 24 h after transfection or induction. In brief, cells were fixed using 4% (wt/vol) or 8% (wt/vol) formaldehyde solution, permeabilized by incubation with a 0.25% (vol/vol) Triton X-100 solution, and blocked with a 5% (wt/vol) BSA solution (Wurm et al, 2010). Proteins of interest were labeled with antibodies against DmMIC10b (this study), FLAG-tag (Merck/Sigma-Aldrich), Spot-tag (ChromoTek), TOMM22 (Merck/Sigma-Aldrich), and MIC60 (Abcam). For STED nanoscopy, detection was achieved via secondary antibodies custom-labeled with Alexa Fluor 594 or STAR RED. Cells were mounted using Mowiol containing 1,4-diazabicyclo[2.2.2]octane (DABCO).

For 4Pi-STORM, samples were prepared on 18-mm-diameter glass coverslips, which were coated over one quarter of their surface with a reflective aluminum layer, which was used for initial alignment of the sample within the 4Pi-STORM microscope. Cells were plated on the coated side of the coverslip and fluorescently labeled as described before (Bates et al, 2022). For two-color imaging, cells were fixed and stained with antibodies against MIC60 (Abcam) and the FLAG epitope (Sigma-Aldrich). Primary antibodies were detected with Fab fragments coupled to Alexa Fluor 647 (Thermo Fisher Scientific) or antibodies labeled with Cy5.5 (Jackson ImmunoResearch). Before imaging, the sample was washed with PBS and mounted in STORM imaging buffer in a sandwich configuration with a second glass coverslip. The two coverslips were sealed together around their edge with fast-curing epoxy glue (UHU GmbH).

## Sample preparation for live-cell STED nanoscopy

HeLa and COS-7 cells were seeded in glass bottom dishes (ibidi GmbH) and grown at 37°C and 5% $CO_2$ overnight. Cells were co-transfected with AAVS1-TRE3G-DmMIC10b-FLAG-T2A-EGFP and AAVS1-Blasticidin-CAG-COX8A-SNAP (Stephan et al, 2019). 4 h after transfection, the culture medium was exchanged and the expression of DmMIC10b-FLAG-T2A-EGFP was induced by adding 1 μg/ml doxycycline hyclate (Sigma-Aldrich) to the growth medium for 24 h. Before STED imaging, cells were stained with 1 μM SNAP-cell

SiR-647 (NEB) for 30 min at 37°C and 5% $CO_2$. Transfected cells were identified based on the cytosolic EGFP reporter and recorded by live-cell STED nanoscopy at RT.

## Sample preparation of cultivated cells for electron tomography

Aclar 33C disks were punched with 18 mm diameter using a 0.198-mm-thick Aclar film (Plano GmbH) and sterilized with 70% ethanol before usage. On these, COS-7 cells or MICOS-depleted HeLa cells (Stephan et al, 2020) were grown to ~70% confluency and transfected with pcDNA3.1-DmMIC10b-FLAG. 24 h after transfection, samples were immobilized and fixed with 2.5% (vol/vol) glutaraldehyde in 0.1 M cacodylate buffer at pH 7.4, first for 1 h at RT, then transferred to 4°C until the next day. Alternatively, S2 cells were seeded on coated glass coverslips and transfected the next day as described above. They were fixed by immersion with 2.5% (vol/vol) glutaraldehyde in 0.1M phosphate buffer (pH 7.4).

Before post-fixation, the cells were in addition stained with 1% osmium tetroxide and 1.5% (wt/vol) $K_4[Fe(CN)_6]$ in 0.1 M cacodylate buffer at pH 7.4 for 1 h at RT. After post-fixation in 1% osmium tetroxide for 1 h at RT and pre-embedding en bloc staining with 1% (wt/vol) uranyl acetate for 30 min at RT, samples were dehydrated and embedded in Agar 100 resin (Plano GmbH).

## Sample preparation of fly tissues for electron microscopy

After extraction, fly brains were fixed in bulk by immersion with 2% (vol/vol) glutaraldehyde in 0.1 M cacodylate buffer at pH 7.4. Fixation was completed overnight at 4°C. After three washing steps with 0.1 M cacodylate buffer, the brains were stained with 1% (wt/vol) osmium tetroxide for 1 h at RT followed by en bloc staining with 1% (wt/vol) uranyl acetate for 30 min at RT. Subsequently, the brains were dehydrated in a graded series of ethanol and finally flat-embedded in a thin layer of Agar 100 resin in between two Aclar sheets. Individual brains were manually cut out of the resin with a saw, and 80-nm-thick sections were collected on Formvar-coated copper grids. 10 × 10 tiles at 8,600x original magnification were recorded at randomly selected areas of the fly brains.

## Preparation of recombinant DmMIC10b for negative stain EM

DmMIC10b purified in the presence of urea was dialyzed (against 20 mM Tris, 100 mM NaCl, and 2 mM DTT, pH 8.0) to precipitate DmMIC10b. The precipitate was solubilized by adding 10% (wt/vol) sarcosyl and 5% (wt/vol) digitonin followed by thorough pipetting. The obtained solution was afterward diluted with a suspension buffer containing 20 mM Tris, 100 mM NaCl, 3% (wt/vol) DDM, and 2mM DTT, pH 8.0, up to a final concentration of 0.2% sarcosyl and 0.05% digitonin. The solution was supplemented with 10 mM imidazole and loaded onto magnetic His-Beads. Samples were rotated on a wheel for 1.5 h at 4°C. The supernatant was removed, and the beads were washed five times with a wash buffer containing 20 mM Tris, 100 mM NaCl, 10 mM imidazole, 1 mM DTT, and 0.05% (wt/vol) DDM, pH 8.0. Afterward, the beads were washed five times with a wash buffer containing 20 mM Tris, 100 mM NaCl, 1 mM imidazole, 1 mM DTT, and 0.01% (wt/vol) DDM, pH 8.0. Elution was performed with 500 mM imidazole. Samples were dialyzed (against 20 mM Tris,

100 mM NaCl, 1 mM DTT, and 0.01% [wt/vol] DDM, pH 8.0) using a 5 kD cutoff membrane. The samples were analyzed by SDS–PAGE and Coomassie brilliant blue staining. For negative staining EM, the obtained protein samples were bound to a glow-discharged carbon foil–covered 400-mesh copper grid. After successive washing with water, samples were stained with 1% (wt/vol) uranyl acetate aq. and evaluated at RT using a Talos L120C transmission electron microscope (Thermo Fisher Scientific).

### Fluorescence microscopy

Imaging of testes was performed on a Zeiss LSM 700 confocal laser-scanning microscope (Carl Zeiss AG). Confocal light microscopy images of cultured cells were acquired using a Leica TCS SP8 confocal laser-scanning microscope (Leica Microsystems) equipped with an HC PL APO 63x/1.40 Oil objective (Leica Microsystems).

STED nanoscopy was performed using an Expert Line quad scanning STED microscope (Abberior Instruments) equipped with a UPlanSApo 100x/1.40 Oil objective (Olympus). Alexa Fluor 594 was excited at a 561-nm wavelength, and STAR RED was excited at a 640-nm wavelength. STED was performed at a 775-nm wavelength. SiR-647 was excited at a 640-nm wavelength, and STED was performed at a 775-nm wavelength as described previously (Stephan et al, 2019, SciRep).

For 4Pi-STORM, cells were imaged on the 4Pi-STORM microscope by illuminating with 642-nm laser light and recording the on–off blinking events of individual fluorescent molecules on a CCD camera, for ~100,000 camera frames at a rate of 100 Hz. To achieve a larger imaging depth, the sample was periodically shifted along the optical axis (z-coordinate) in steps of 500 nm during the recording. Afterward, in a post-processing step, each blinking event was analyzed to determine a 3D molecular coordinate, and the full set of coordinates plotted together in 3D space forms the 3D 4Pi-STORM image. Multicolor imaging was achieved by distinguishing the Alexa Fluor 647 and Cy5.5 according to the ratio of photons for each fluorophore detected in the four image channels of the microscope. The spatial resolution of the processed 4Pi-STORM image is ~10 nm in all dimensions. Full details of the microscope, image procedure, and data analysis are described in Bates et al (2022).

### Electron microscopy

For 2D analysis, images of ultrathin sections of ~70 nm thickness were recorded on a Philips CM120 BioTwin transmission electron microscope (Philips Inc.). Usually, 2D images of at least 20 different cells were randomly recorded for each sample at 8,600× original magnification using a TemCam 224A slow-scan CCD camera (TVIPS).

For electron tomography, tilt-series of thin sections of ~230 nm that were in addition decorated with 10-nm gold beads on both sides were recorded on a Talos L120C transmission electron microscope (Thermo Fisher Scientific/FEI Company) at 17,500× original magnification using a Ceta 4k × 4k CMOS camera in an unbinning mode. Series were recorded from −65.0° to 65.0° with a 2° dose-symmetric angular increase. The series were calculated using Etomo (David Mastronade; http://bio3d.colorado.edu/).

### Imaging data processing

4Pi-STORM data were analyzed using the custom MATLAB code. Single X,Y,Z coordinates were extracted from raw HDF5 files generated by STORM acquisition software, and split into two-channel datasets based on logical values embedded in the data structure. Nearest neighbor distances (minimum pairwise distances) were then calculated between the two channels, and complete pairwise distances were calculated within each channel, using inbuilt MATLAB functions. For the complete pairwise self-distances for each channel, a global threshold distance of 200 nm was used to filter the data, in order to probe the differences at distances comparable to mitochondrial diameters. The data were then plotted as empirical cumulative distribution functions and displayed on plots comparing the different conditions in the context of MIC10 expression levels. For volume renderings of 4Pi-STORM data, we relied on Imaris (Bitplane).

TEM recordings of thin sections and ET data were processed in Fiji using the median filter. For the analysis of cristae morphology, the images were manually categorized in a blinded approach based on the ultrastructure of individual mitochondrial cross sections. For automated segmentation of mitochondrial membranes, we employed MemBrain-Seg (https://github.com/teamtomo/membrain-seg) using the current provided model V10 and a sliding window size of 128. The membrane model was further refined manually in Amira 6.0.1 (Thermo Fisher Scientific). Filaments were manually segmented in IMOD 4.11 and imported into Amira for visualization. If not stated otherwise, STED nanoscopy data were smoothed with a low-pass filter using Imspector software (Abberior Instruments). For deconvolution, we used the Richardson Lucy algorithm in Imspector software. Confocal images of the fly tissue were processed using ImageJ and Adobe Illustrator. The seminal vesicle areas were calculated using the "Measure" feature in ImageJ.

## Data Availability

There are no primary datasets and computer codes attached to this article. All data reported in this study will be shared by the lead contact upon reasonable request.

## Supplementary Information

## Acknowledgements

We thank Sylvia Loebermann, Rita Schmitz-Salue, Karin Hartwig, and Doris Brentrup for excellent technical assistance and maintenance of fly stocks. This work was supported by the European Research Council (ERCAdG No. 835102, to S Jakobs). It was funded by the DFG-funded FOR2848 (project P05 to M Meinecke and Z01 to D Riedel) and SFB1190 (project P01 to S Jakobs and

P13 to P Rehling). Stocks obtained from the Bloomington Drosophila Stock Center (NIH P40OD018537) were used in this study.

## Author Contributions

T Stephan: conceptualization, data curation, formal analysis, investigation, visualization, methodology, and writing—original draft, review, and editing.

S Stoldt: conceptualization, data curation, formal analysis, validation, investigation, visualization, methodology, and writing—original draft, review, and editing.

M Barbot: conceptualization, data curation, validation, investigation, visualization, methodology, and writing—review and editing.

TD Carney: validation, investigation, methodology, and writing—review and editing.

F Lange: validation, investigation, and writing—review and editing.

M Bates: software, investigation, visualization, methodology, and writing—review and editing.

P Bou Dib: validation, investigation, methodology, and writing—review and editing.

K Inamdar: software, formal analysis, supervision, validation, visualization, methodology, and writing—review and editing.

HR Shcherbata: resources, supervision, validation, and writing—review and editing.

M Meinecke: resources, validation, investigation, visualization, methodology, and writing—review and editing.

D Riedel: conceptualization, validation, investigation, visualization, methodology, and writing—review and editing.

S Dennerlein: conceptualization, supervision, validation, investigation, methodology, and writing—review and editing.

P Rehling: conceptualization, data curation, formal analysis, supervision, funding acquisition, validation, project administration, and writing—original draft, review, and editing.

S Jakobs: conceptualization, data curation, software, formal analysis, supervision, funding acquisition, validation, visualization, methodology, project administration, and writing—original draft, review, and editing.

## Conflict of Interest Statement

The authors declare that they have no conflict of interest.

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
