## [Reviewer comments · Life Science Alliance]

Life Science Alliance

Drosophila MIC10b can polymerize into cristae-shaping filaments

Till Stephan, Stefan Stoldt, Mariam Barbot, Travis Carney, Felix Lange, Mark Bates, Peter Bou Dib, Kaushik Inamdar, Halyna Shcherbata, Michael Meinecke, Dietmar Riedel, Sven Dennerlein, Peter Rehling, and Stefan Jakobs

DOI: <https://doi.org/10.26508/lsa.202302177>

Corresponding author(s): Stefan Jakobs, Max Planck Institute for Multidisciplinary Sciences

Review Timeline:

Submission Date:	2023-05-23
Editorial Decision:	2023-06-22
Revision Received:	2023-11-28
Editorial Decision:	2023-12-18
Revision Received:	2024-01-08
Accepted:	2024-01-09

Transaction Report:

June 22, 2023

Re: Life Science Alliance manuscript #LSA-2023-02177-T

Prof. Stefan Jakobs
Max-Planck-Institute for Biophysical Chemistry
Department of NanoBiophotonics/Mitochondrial Structure and Dynamics
Am Fassberg 11
Goettingen 37077
Germany

Dear Dr. Jakobs,

Thank you for submitting your manuscript entitled "The Drosophila MIC10 orthologue has a propensity to polymerize into cristae-shaping filaments" to Life Science Alliance. The manuscript was assessed by expert reviewers, whose comments are appended to this letter. We invite you to submit a revised manuscript addressing the Reviewer comments.

Thank you for this interesting contribution to Life Science Alliance. We are looking forward to receiving your revised manuscript.

Sincerely,

B. MANUSCRIPT ORGANIZATION AND FORMATTING:

Reviewer #1 (Comments to the Authors (Required)):

The Drosophila MIC10 orthologue has a propensity to polymerize into cristae-shaping filaments

The introduction is well written and provides sufficient background to indicate the differences in MICOS between organisms and therefore emphasizes the increase in knowledge that the investigations into Drosophila have brought to the field.

Methods are comprehensive, accurately described in adequate detail for repetition of the experiments.

There is a wealth of data figures, which are all of high quality, accompanied by informative and accurate legends.

The ability of DmMIC10b to form filaments independently of other members of the MICOS complex is striking and more so that it can do this in vitro but that this is not a preserved characteristic in yeast or human MIC10.

The video loops clearly show the 3D nature and spatial distribution of DmMIC10b vs MIC60. The illustration of this is appropriate as supplemental material.

Data is accurately interpreted with no over-reaching conclusions.

In summary, this constitutes a very thorough, extensive and exceedingly well carried out investigation and analysis of the functions of the MIC10 orthologues in Drosophila. It is highly worthy of publication with no changes.

Minor points

- The oligo sequences in the tables should indicate that these are all given as 5'->3'
- Line 285 - Sepharose should start with a capital letter

Reviewer #2 (Comments to the Authors (Required)):

In this manuscript, Stephan and colleagues characterize one of the Drosophila forms of MIC10 (CG41128). They demonstrate that the ubiquitously expressed CG41128 is the Mic10 homolog and is required for Drosophila development. In cell overexpression studies they observe that DmMIC10b has an interesting propensity to form filaments and cause drastic changes to mitochondrial shape. They use this assay to confirm the importance of glycine repeats in polymer formation (as previously shown for yeast Mic10) and further suggest that a Cys residue is required to filament. Suppression of Mic10 filaments could be achieved by coexpressing other MICOS subunits.

This study presents interesting new observations, and the experiments are largely clear, however the investigation into filament formation due to large overexpression leaves me wondering about the significance of this finding. Aspects to improve the manuscript are below.

- a. The image analysis comparing CG41128 knockout cristae morphology to known MIC10 knockout cristae morphology is a good comparison however descriptions of how the quantitation of 'altered cristae' is lacking.
- b. Line 510 suggests that CG41128 is the "major" MIC10 ortholog, but has not been compared to the other two MIC10 ortholog introduced previously. It may be necessary to change the wording here to suggest that this is a major ortholog, but it cannot be certain that the other two orthologs are not equally important.
- c. Fig 3: A co-stain to confirm the specific localisation of the overexpressed CG41128 could help this section to confirm that either all of the signal is mitochondrial, or that maybe not all of the signal is mitochondrial. A mitochondrial co-stain such as TOM20 may help, especially with supporting claims that MIC60 is at the IBM as currently there is no spatial reference point to compare to.
- d. When discussing that the filamentous bundles appear to not be at the IBM, are they outside the IBM, and outside the mitochondria? Or are they outside the IBM, but in the matrix? This could be clarified more.
- e. Line 671 - This should refer to figure 6C not 6B. And on line 673
- f. Fig 6: the authors should validate that the mutant dMic10 forms are expressed at similar levels and are not degraded using western blotting.

Reviewer #3 (Comments to the Authors (Required)):

In the manuscript "The *Drosophila* MIC10 orthologue has a propensity to polymerize into cristae-shaping filaments", the authors describe DmMIC10b as the major MIC10 orthologue out of 3 MIC10 like proteins in *Drosophila melanogaster*, and further show some interesting and unique features of this protein. While DmMIC10b shares similarities with other well studied MIC10 proteins, such as the predicted structure, conserved glycine-rich motifs, cristae shaping functions, and ability of oligomerization, DmMIC10b seems to be the only MIC10 like protein identified so far that can form mitochondrial morphology-shaping filaments. The molecular basis of DmMIC10b was further investigated by point mutations. Altogether, this is a very interesting and thought-provoking study since it not only describes the major *Drosophila* MIC10 orthologue and its novel filament-forming ability, but also draws attention to the molecular mechanisms of protein homo-oligomerization as well as to the diversity of the MICOS complex from different organisms. The nanoscopic methods are highly advanced and yield stunning results. Overall this is an excellent study. A few open questions remain to understand the mechanism of DmMIC10b behaviour that can be addressed in the context of a minor revision.

Major points:

1. The submitochondrial localization of DmMIC10b filaments remains vague and should be addressed directly. It would be especially convincing to address this question in S2 cells rather than mammalian cells. The authors describe the filaments as being between IBM and OM, suggesting they consist of non-membrane integral Mic10. Despite this, the IM is drastically reshaped upon filament formation. Is this because (as the authors state) filaments can be also inside the matrix? The authors should investigate membrane integration and submitochondrial localization for example by biochemical methods.
2. The mutational analysis demonstrates clearly that oligomerization via the second TM segment as well as the Cys19 are required for filament formation. However, since both these elements are widely conserved, neither of them explains this unique property of DmMIC10b. In contrast, the glycine motif in the first TM segment does appear to be unique: Instead of the metazoan Gx(G)xGx(G)x motif, where Gly residues (mostly) alternate with large hydrophobic ones, the sequence in DmMIC10b is GCG(G)VII(G)S (with the conserved Gly residues in brackets). This conceivably alters the mode of interactions via the first TM segment quite significantly. It would be worth testing whether mutations that disrupt these extended patches of Gly or bulky residues / introduce partial changes similar to the mammalian motif interfere with filament formation.

Minor points:

1. The authors show that DmMIC10b can interact with orthologous MICOS components in mammalian cells. Can the protein also complement absence of the endogenous MIC10 phenotypically?
2. In Figure 3, the authors show that when moderately expressed, DmMIC10b forms puncta. Are these localized at crista junctions, do they co-localize with Mic60?

Point by point response to the reviewers' concerns

We thank all reviewers for evaluating our work and for their valuable comments and constructive suggestions. We have revised the manuscript according to the raised points by rewriting parts of the results. We followed the reviewers' suggestions and updated Figure 3 and Figure 5. Moreover, we extended Supplementary Figures S4, S9, S10 and S11 and added substantial new data. Finally, we added two entirely new Figures, namely Supplementary Figures S5 and S8 to better illustrate the submitochondrial location of DmMIC10b filaments. We believe that by addressing the referees' concerns thoroughly, we were able to significantly improve our manuscript and that our work is now suitable for publication. In the following, we will address each comment raised by the reviewers point by point.

Reviewer 1

The introduction is well written and provides sufficient background to indicate the differences in MICOS between organisms and therefore emphasizes the increase in knowledge that the investigations into Drosophila have brought to the field. Methods are comprehensive, accurately described in adequate detail for repetition of the experiments. There is a wealth of data figures, which are all of high quality, accompanied by informative and accurate legends. The ability of DmMIC10b to form filaments independently of other members of the MICOS complex is striking and more so that it can do this in vitro but that this is not a preserved characteristic in yeast or human MIC10. The video loops clearly show the 3D nature and spatial distribution of DmMIC10b vs MIC60. The illustration of this is appropriate as supplemental material. Data is accurately interpreted with no over-reaching conclusions. In summary, this constitutes a very thorough, extensive and exceedingly well carried out investigation and analysis of the functions of the MIC10 orthologues in Drosophila. It is highly worthy of publication with no changes.

We thank Referee 1 for the positive assessment of our study.

Minor points

- The oligo sequences in the tables should indicate that these are all given as 5'→3'
- Line 285 - Sepharose should start with a capital letter

Done.

Reviewer 2

In this manuscript, Stephan and colleagues characterize one of the Drosophila forms of MIC10 (CG41128). They demonstrate that the ubiquitously expressed CG41128 is the Mic10 homolog and is required for Drosophila development. In cell overexpression studies they observe that DmMIC10b has an interesting propensity to form filaments and cause drastic changes to mitochondrial shape. They use this assay to confirm the importance of glycine repeats in polymer formation (as previously shown for yeast Mic10) and further suggest that a Cys residue is required to filament. Suppression of Mic10 filaments could be achieved by coexpressing other

MICOS subunits. This study presents interesting new observations, and the experiments are largely clear, however the investigation into filament formation due to large overexpression leaves me wondering about the significance of this finding. Aspects to improve the manuscript are below.

We thank Referee 2 for the positive comments on our work. We agree that filament formation of DmMIC10 can be induced by overexpression. However, as DmMIC13, but not HsMIC13, efficiently prevents the formation of filaments, we are convinced that the proteins' propensity to polymerize is highly related to its structural function in fly mitochondria.

In the discussion section of the revised manuscript, we highlight the fact that the biological function of DmMIC10b polymerization remains to be explored in future studies. We note in the discussion section that DmMIC10b filaments are reminiscent of filaments formed by GFP-tagged variants of the bacterial cytoskeleton protein Mreb. Those long Mreb filaments are considered a tagging/overexpression artifact, but their formation is strongly related to the proteins' function. Instead of forming long filaments, untagged Mreb polymerizes into rings which are crucial for the division of bacterial cells. We suggest that the formation of DmMIC10b filaments indicates that DmMIC10b tends to polymerize into well-organized structures to control the curvature of the membrane and that the formation of these structures is fine-tuned by other MICOS subunits.

The image analysis comparing CG41128 knockout cristae morphology to known MIC10 knockout cristae morphology is a good comparison however descriptions of how the quantitation of 'altered cristae' is lacking.

The images were manually categorized in a blinded approach. We added a paragraph describing this approach to the "Material and Methods" section. See "Imaging data processing".

Line 510 suggests that CG41128 is the "major" MIC10 ortholog, but has not been compared to the other two MIC10 ortholog introduced previously. It may be necessary to change the wording here to suggest that this is a major ortholog, but it cannot be certain that the other two orthologs are not equally important.

The referee is correct that our study does not investigate the role of the other two MIC10-like proteins in flies. However, expression data accessible via the FlyBase data base show clearly that DmMIC10b is the only protein out of the three putative orthologues that is expressed in all tissues and throughout all developmental stages. Hence, MINOS1a and MINOS1c might be tissue specific MIC10s, although their interaction with MICOS has, to our best knowledge, not yet been confirmed. We do not exclude the possibility that these two proteins play a significant role during fly development. Nevertheless, we want to point out that also DmMIC10b is also present in testes (Supplementary Figure S2). In line with this, loss of DmMIC10b reduced fertility of flies, underlining its importance. In order to address the referees' concerns, we have partially rewritten the respective section. It now reads:

“Altogether, the ubiquitously expressed Dmel_CG41128 shares sequence homology with other MIC10 proteins, presumably features a hairpin-like structure and binds to the MICOS complexes of D. melanogaster, H. sapiens and C. aethiops. Dmel_CG41128 further regulates the levels of DmMIC13 in flies, and its depletion strongly affects the cristae architecture. These findings support the view that Dmel_CG41128 is the ubiquitously expressed MIC10 orthologue in D. melanogaster. Hence, in accordance with the uniform nomenclature for MICOS (Pfanner et al., 2014), we will refer to it as DmMIC10b from here on.”

Fig 3: A co-stain to confirm the specific localization of the overexpressed CG41128 could help this section to confirm that either all of the signal is mitochondrial, or that maybe not all of the signal is mitochondrial. A mitochondrial co-stain such as TOM20 may help, especially with supporting claims that MIC60 is at the IBM as currently there is no spatial reference point to compare to.

We thank the Referee for this valuable suggestion. As we could not find an antibody that achieves sufficient labeling of the outer membrane in fly cells, we co-labeled DmMIC10b together with DmMIC60 in S2 cells. In addition, we added dual-color STED images of DmMIC10b/MIC60 or DmMIC10b/TOM20 in COS-7 cells. The microscopy images, which are now presented in Figure 3C and Supplementary Figure S4B, show clearly that in both cell types the filaments form exclusively inside mitochondria. In the manuscript it now reads:

“Intriguingly, long DmMIC10b-containing filaments, often forming bundles of filaments, were observable at higher expression levels of DmMIC10b-FLAG (Figure 3A, B). These DmMIC10b-containing filaments seemed to be located inside of mitochondria and pervaded the mitochondrial network (Figure 3A-C).”

And:

“To explore this idea, we first expressed DmMIC10b-FLAG in two heterologous cellular systems, namely HeLa and COS-7 cells. Similar to the overexpression in S2 cells, expression of DmMIC10b in these cells resulted in the formation of DmMIC10b-containing filaments, which formed exclusively inside mitochondria (Figure 4A, Supplementary Figure S4).”

When discussing that the filamentous bundles appear to not be at the IBM, are they outside the IBM, and outside the mitochondria? Or are they outside the IBM, but in the matrix? This could be clarified more.

We apologize for the confusion. We have made significant changes to Figure 5 and the results section to better explain the submitochondrial localization of the filaments. We segmented the membranes and some of the filaments in the ET data set shown in Figure 5. Moreover, we have added the raw data to Supplementary Figure S8A and colored the entire inter membrane space (crista lumen as well as the space between the OM and the IBM) to further highlight the position of the filaments. The figure legend to Figure 5 now reads:

“Figure 5: DmMIC10b polymerizes into filaments which alter the mitochondrial ultrastructure. A. Live-cell STED of mammalian cells expressing DmMIC10b. COS-7 and HeLa cells were co-transfected with DmMIC10b-FLAG and COX8A-SNAP to visualize the cristae membrane. Cells were labeled with SNAP-cell SiR 647 and visualized by 2D live-cell STED nanoscopy. B. Electron tomography of mitochondria from COS-7 cells expressing DmMIC10b-FLAG. The recording shows a cross-section (mito 1) and a longitudinal section (mito 2) of two adjacent mitochondria. Membranes (blue) and filaments (yellow) were semiautomatically segmented and are shown as volume renderings. Abbreviations: Outer membrane (OM), inner boundary membranes (IBM), cristae membrane (CM). Scale bars: 2 μ m (A); 0.25 μ m (B).”

In addition, we recorded ET data of DmMIC10b filaments in S2 cells, which we now also show in Supplementary Figure S8. In both cell types, filaments appeared in the entire inter membrane space. If these filaments are surrounded by a lipid bilayer, or if the hydrophobic amino acids are shielded inside the filaments remains unclear and will most likely require structural biology approaches to address. (See also comment to Referee 3).

Line 671 - This should refer to figure 6C not 6B. And on line 673

Thank you, we corrected this mistake.

Fig 6: the authors should validate that the mutant dMic10 forms are expressed at similar levels and are not degraded using western blotting.

We thank the referee for this excellent suggestion. We added the respective western blot analysis to Supplementary Figure S11. It shows that the expression of the C19S mutant, which is incapable of forming filaments, is comparable to that of WT DmMIC10b. In contrast, the C28S mutant which still forms filaments, shows a significantly lower expression. Therefore, the data prove that the effect by C19S substitution is unrelated to expression levels.

Reviewer 3

In the manuscript "The Drosophila MIC10 orthologue has a propensity to polymerize into cristae-shaping filaments", the authors describe DmMIC10b as the major MIC10 orthologue out of 3 MIC10 like proteins in Drosophila melanogaster, and further show some interesting and unique features of this protein. While DmMIC10b shares similarities with other well studied MIC10 proteins, such as the predicted structure, conserved glycine-rich motifs, cristae shaping functions, and ability of oligomerization, DmMIC10b seems to be the only MIC10 like protein identified so far that can form mitochondrial morphology-shaping filaments. The molecular basis of DmMIC10b was further investigated by point mutations. Altogether, this is a very interesting and thought-provoking study since it not only describes the major Drosophila MIC10 orthologue and its novel filament-forming ability, but also draws attention to the molecular mechanisms of protein homo-oligomerization as well as to the diversity of the MICOS complex from different organisms. The nanoscopic methods are highly advanced and

yield stunning results. Overall, this is an excellent study. A few open questions remain to understand the mechanism of DmMIC10b behaviour that can be addressed in the context of a minor revision.

We thank Referee 3 for the positive comments on our manuscript.

Major points:

1. The submitochondrial localization of DmMIC10b filaments remains vague and should be addressed directly. It would be especially convincing to address this question in S2 cells rather than mammalian cells. The authors describe the filaments as being between IBM and OM, suggesting they consist of non-membrane integral Mic10. Despite this, the IM is drastically reshaped upon filament formation. Is this because (as the authors state) filaments can be also inside the matrix? The authors should investigate membrane integration and submitochondrial localization for example by biochemical methods.

We followed the referees' suggestion and further analyzed the localization of DmMIC10b in mitochondria of S2 cells by electron tomography (Supplementary Figure S8) and by carbonate extraction (Referee Figure 1). Tomograms showed that both in COS-7 and S2 cells, filaments were found outside of the contiguous inner membrane. The filaments resided in the inter membrane space; i. e. we found them in the crista lumen as well as between the OM and the IBM. To better illustrate the localization of the filaments, we now added a segmentation of membranes and filaments to Figure 5. It now reads:

“Figure 5: DmMIC10b polymerizes into filaments which alter the mitochondrial ultrastructure. A. Live-cell STED of mammalian cells expressing DmMIC10b. COS-7 and HeLa cells were co-transfected with DmMIC10b-FLAG and COX8A-SNAP to visualize the cristae membrane. Cells were labeled with SNAP-cell SiR 647 and visualized by 2D live-cell STED nanoscopy. B. Electron tomography of mitochondria from COS-7 cells expressing DmMIC10b-FLAG. The recording shows a cross-section (mito 1) and a longitudinal section (mito 2) of two adjacent mitochondria. Membranes (blue) and filaments (yellow) were semiautomatically segmented and are shown as volume renderings. Abbreviations: Outer membrane (OM), inner boundary membranes (IBM), cristae membrane (CM). Scale bars: 2 μm (A); 0.25 μm (B).”

Our tomograms repeatedly showed DmMIC10b-filaments forming in the inter membrane space. However, our recordings do not allow to answer the question whether these filaments are surrounded by a membrane shell. Theoretically, the hydrophobic amino acids of DmMIC10b could be shielded inside the filaments, resulting in filaments without a lipid shell within the aqueous IMS. Alternatively, the filaments might be surrounded by a lipid shell. In this case, the filament-forming DmMIC10b molecules would be membrane embedded. When performing carbonate extractions, DmMIC10b-FLAG behaved similar to DmMIC60 (See Referee Figure 1), which may support the idea that a fraction of the filaments is membrane-embedded or membrane-attached. However, since the carbonate extraction cannot distinguish between proteins that are insoluble and those that are retained in the membrane fraction, because they are transmembrane proteins, we decided to exclude the data from the manuscript. We

believe that answering this question with confidence will require *in situ* electron cryotomography, which can visualize lipid bilayers but is outside of the scope of this study.

Referee Figure 1: Carbonate extraction from mitochondria of S2 cells. S2 cells were transfected to overexpress DmMIC10b-FLAG. Mitochondria were isolated and analyzed by carbonate extraction.

2. The mutational analysis demonstrates clearly that oligomerization via the second TM segment as well as the Cys19 are required for filament formation. However, since both these elements are widely conserved, neither of them explains this unique property of DmMIC10b. In contrast, the glycine motif in the first TM segment does appear to be unique: Instead of the metazoan Gx(G)xGx(G)x motif, where Gly residues (mostly) alternate with large hydrophobic ones, the sequence in DmMIC10b is GCG(G)VII(G)S (with the conserved Gly residues in brackets). This conceivably alters the mode of interactions via the first TM segment quite significantly. It would be worth testing whether mutations that disrupt these extended patches of Gly or bulky residues / introduce partial changes similar to the mammalian motif interfere with filament formation.

We thank the referee for this valuable suggestion. We tested a variety of different additional mutants and included them in Supplementary Figure S10. Moreover, we have rewritten large parts of the results section to better explain the role of the glycine-rich motifs.

In short, we found that filament formation can be prevented by exchanging the entire first glycine-rich motif with the one from human MIC10. However, this change also lowered the expression levels of the protein quite significantly (Supplementary Figure S10C). Point mutations affecting one or two glycine residues that are part of the core GxxxG motif also blocked the formation of filaments, whereas a substitution outside of the core GxxxG motif (G29L) did not interfere with the formation of filaments.

The referee is correct that both C19 and the GxGxGxG motif in the second TMD are highly conserved across different species. We rigorously tried to identify all amino acids that might cause the different behavior of HsMIC10 and DmMIC10b. As described above, our data may support the idea that the first TMD is also involved in filament formation, but facing the differences in expression levels, we do not want to overinterpret our data. We believe that structural biology will be inevitable to fully understand the structural basis of the oligomerization/ polymerization.

Minor points:

1. The authors show that DmMIC10b can interact with orthologous MICOS components in

mammalian cells. Can the protein also complement absence of the endogenous MIC10 phenotypically?

To answer this question, we updated Supplementary Figure S9 by adding an analysis concerning the cristae morphology. DmMIC10b cannot rescue the aberrant cristae morphology in human mitochondria devoid of HsMIC10. In line with this, we observed that DmMIC13 but not HsMIC13 can prevent the formation of DmMIC10b filaments, suggesting that although DmMIC10b can interact with mammalian MICOS, it may not be able to support inner membrane shaping due to deficient regulation of MIC10 oligomer formation. It now reads:

“In MIC60-KO cells, we observed the majority of filaments in bundles between OM and IBM (Supplementary Figure S9A). Despite its ability to interact with human MICOS, expression of DmMIC10b could not rescue the aberrant cristae morphology in human MIC10-KO cells (Supplementary Figure S9A-C). Instead, we observed the formation of bundles of filaments that remodeled the cristae membrane in a similar way as observed in WT cells (Supplementary Figure S9D).”

2. In Figure 3, the authors show that when moderately expressed, DmMIC10b forms puncta. Are these localized at crista junctions, do they co-localize with Mic60?

We thank the referee for this question. We performed a nearest neighbor analysis using our 4Pi STORM recordings and included this into the manuscript as Supplementary Figure S5. The analysis shows that at low expression, DmMIC10b colocalized with MIC60, indicating that it was present in the IBM. Once the levels increased DmMIC10b could also be found inside the mitochondria, suggesting that it moved into the cristae membranes. Finally, with strong overexpression, extended filaments were formed. The majority of these filaments seemed to be devoid of MIC60. In the paper it now reads:

“To localize DmMIC10b in 3D, we next performed 4Pi-STORM of COS-7 cells expressing DmMIC10b-FLAG (Bates et al., 2022). At low expression levels, DmMIC10b was found in close proximity to MIC60 clusters, suggesting that it was mainly located at CJs or in the IBM (Supplementary Figure S5). However, the 3D recordings confirmed that at higher expression levels, a large fraction of DmMIC10b was present in filamentous structures (Supplementary Movie S1). Most of these long filaments were located along the center of mitochondrial tubules with no obvious connection to the MIC60 clusters indicating the IBM (Figure 4B, Supplementary Figure S5 and Supplementary Movie S2).”

December 18, 2023

RE: Life Science Alliance Manuscript #LSA-2023-02177-TR

Prof. Stefan Jakobs
Max-Planck-Institute for Biophysical Chemistry
Department of NanoBiophotonics/Mitochondrial Structure and Dynamics
Am Fassberg 11
Goettingen 37077
Germany

Dear Dr. Jakobs,

Thank you for submitting your revised manuscript entitled "Drosophila MIC10b can polymerize into cristae-shaping filaments". We would be happy to publish your paper in Life Science Alliance pending final revisions necessary to meet our formatting guidelines.

- please upload your main and supplementary figures as single files
- please upload your main manuscript text as an editable doc file
- please add the Twitter handle of your host institute/organization as well as your own or/and one of the authors in our system
- please consult our manuscript preparation guidelines <https://www.life-science-alliance.org/manuscript-prep> and make sure your manuscript sections are in the correct order
- Tables should be numbered consecutively with Arabic numerals (1, 2, 3, 4). They can be included at the bottom of the main manuscript file or sent as separate files. Please be sure to cite them in the manuscript text
- please remove figures and their legends from the manuscript file. They should be uploaded only separately
- please add your main, supplementary figure, movie, and table legends to the main manuscript text after the references section
- please add callouts for Figures S3A-H; S4A-B; S5A-C; S7A-C; S8A-B; S11A-B to your main manuscript text

Figure Checks:

- please add sizes next to the blots in Figure 2, S10 and S11

A. FINAL FILES:

B. MANUSCRIPT ORGANIZATION AND FORMATTING:

Sincerely,

Reviewer #2 (Comments to the Authors (Required)):

I am happy with the response and feel that the manuscript is acceptable for publication

Reviewer #3 (Comments to the Authors (Required)):

The authors have performed admirable work with this revision and addressed all points in a comprehensive and careful manner. This is an excellent study.

January 9, 2024

RE: Life Science Alliance Manuscript #LSA-2023-02177-TRR

Prof. Stefan Jakobs
Max Planck Institute for Multidisciplinary Sciences
Department of NanoBiophotonics/Mitochondrial Structure and Dynamics
Am Fassberg 11
Goettingen 37077
Germany

Dear Dr. Jakobs,

Thank you for submitting your Research Article entitled "Drosophila MIC10b can polymerize into cristae-shaping filaments". It is a pleasure to let you know that your manuscript is now accepted for publication in Life Science Alliance. Congratulations on this interesting work.

DISTRIBUTION OF MATERIALS:

Again, congratulations on a very nice paper. I hope you found the review process to be constructive and are pleased with how the manuscript was handled editorially. We look forward to future exciting submissions from your lab.

Sincerely,
